# Fine-Grained Implicit Sentiment in Financial News: Uncovering Hidden Bulls and Bears

**Gilles Jacobs \*** and **Véronique Hoste**

Language and Translation Technology Team (LT3), Department of Translation, Interpreting and Communication, Ghent University, Groot-Brittaniëlaan 45, 9000 Ghent, Belgium; veronique.hoste@ugent.be
\* Correspondence: gilles.jacobs@ugent.be

**Abstract:** The field of sentiment analysis is currently dominated by the detection of attitudes in lexically explicit texts such as user reviews and social media posts. In objective text genres such as economic news, indirect expressions of sentiment are common. Here, a positive or negative attitude toward an entity must be inferred from connotational or real-world knowledge. To capture all expressions of subjectivity, a need exists for fine-grained resources and approaches for implicit sentiment analysis. We present the SENTiVENT corpus of English business news that contains token-level annotations for target spans, polar spans, and implicit polarity (positive, negative, or neutral investor sentiment, respectively). We both directly annotate polar expressions and induce them from existing schema-based event annotations to obtain event-implied implicit sentiment tuples. This results in a large dataset of 12,400 sentiment–target tuples in 288 fully annotated articles. We validate the created resource with an inter-annotator agreement study and a series of coarse- to fine-grained supervised deep-representation-learning experiments. Agreement scores show that our annotations are of substantial quality. The coarse-grained experiments involve classifying the positive, negative, and neutral polarity of known polar expressions and, in clause-based experiments, the detection of positive, negative, neutral, and no-polarity clauses. The gold coarse-grained experiments obtain decent performance (76% accuracy and 63% macro-F1) and clause-based detection shows decreased performance (65% accuracy and 57% macro-F1) with the confusion of neutral and no-polarity. The coarse-grained results demonstrate the feasibility of implicit polarity classification as operationalized in our dataset. In the fine-grained experiments, we apply the grid tagging scheme unified model for <polar span, target span, polarity> triplet extraction, which obtains state-of-the-art performance on explicit sentiment in user reviews. We observe a drop in performance on our implicit sentiment corpus compared to the explicit benchmark (22% vs. 76% F1). We find that the current models for explicit sentiment are not directly portable to our implicit task: the larger lexical variety within implicit opinion expressions causes lexical data scarcity. We identify common errors and discuss several recommendations for implicit fine-grained sentiment analysis. Data and source code are available.

**Keywords:** target-based sentiment analysis; implicit polarity on schema-based events; investor sentiment detection; company-specific information extraction for financial markets and economics

## 1. Introduction

The increasing availability of large volumes of digital text in the past decades has led to a boost in research in information extraction (IE), a branch of natural language processing (NLP) aiming to obtain structured information from unstructured text. Opinion-mining or sentiment analysis is a subdomains of IE concerned with subjective expressions, which has also thrived in recent years, not in the least due to its huge application potential in marketing and CRM [1–4]. Sentiment analysis (SA) aims to automatically identify "people's opinions, sentiments, evaluations, appraisals, attitudes, and emotions towards entities such as products, services, organizations, individuals, issues, events, topics, and

their attributes" [5]. The value and direction of subjective opinion expressed is termed polarity and is often labeled as positive , neutral, or negative attitude. The dominant line of research in SA focuses on user-generated content such as product reviews and social media posts. The main communicative goal of these text genres is to convey a person's opinion and subjective experience. As such, user-generated text typically conveys opinionated or evaluative content explicitly using words that directly denote subjectivity, emotion, or opinion. As a result, most sentiment analysis research is directed at the detection of subjective words and phrases. However, previous work has shown that objective-oriented texts such as news articles can also express sentiment, potentially in an indirect manner [5–8]. In the implicit case, readers must infer positive or negative impressions expressed by the author through common-sense connotations and world knowledge.

(a)  "The **bad news** is that AbbVie's **top-notch** dividend might **not be safe**."
    → **Explicit**: Author explicitly states their negative attitude with words "bad news" and "not safe". Positive attitude is made explicit by "top-notch".
(b)  "Boeing **stock climbed** on news that it would **increase its production** of 787 aircraft."
    → **Implicit**: Reader must infer a positive connotation through world knowledge.

For instance in example (a), "*bad news*" and "*not be safe*" are phrases that explicitly convey negative consequences for the company AbbVie and its dividend, while "*top-notch*" denotes a positive evaluation. The subjective attitude stems from the inherent semantics of these word and is thus lexically explicit. Example (b) describes events such as "*stock climbing*" and "*increase [in] production*" from which a reader has to infer through real-world knowledge of economics and financial markets that this is positive for the target company (Boeing). In line with previous research on implicit sentiment [6,9], we term any span of words that expresses implicit or explicit sentiment a 'polar span', also called 'sentiment expression' or 'opinion term'. To account for the full range of subjective information contained in text, researchers have started to analyze implicit methods of expressing sentiment [8,10–13]. Nevertheless, the amount of work and resources in the field of implicit SA remains limited and focused on user-generated content, which is a more explicitly opinionated genre. Implicit sentiment is usually treated by coarse-grained methods, with polarity labels created at the document or sentence level, and the field is lacking in substantial gold-standard, human-annotated, token-level datasets. However, the presence of implicit sentiment in economic news [6] combined with potential financial market applications have made it a prime target for implicit sentiment processing.

This paper presents the novel SENTiVENT resource for the processing of fine-grained implicit sentiment in English economic text. In a previous stage, the SENTiVENT corpus was annotated with schema-based event annotations [14]. These event schemata denote an event of a certain type (e.g., product releases, revenue increases, or security value movements, deals) and relate which participating entities play a role in the event (e.g., a product, the amount of increase in stock price, and the main companies involved in a deal). We propose a method for inducing implicitly subjective sentiment–target relationships from fine-grained event schemata, alongside directly annotated expressions of sentiment and their targets. We validate the created resource in an inter-annotator agreement (IAA) study and a set of pilot experiments: coarse-grained implicit polarity classification for gold and clause-based polar expressions, and fine-grained extraction of triplet experiments of <polar span, target span, polarity> triplets. The main contributions of this work can be summarized as follows:

- We construct a corpus with gold-standard annotations for economic sentiment as implicit polar expressions. The domain of implicit SA is lacking fine-grained, target-based datasets and our SENTiVENT dataset fills that gap with a substantially-sized resource containing 12,400 sentiment tuples. We use the event schemata as a basis for targeted implicit sentiment analysis, demonstrating efficient resource creation in fine-grained information extraction. Additionally, separate sentiment–target annotations are made. The quality of all annotations is demonstrated in an IAA study.

- We validate implicit sentiment with coarse-grained pilot experiments by polarity classification of gold polar expressions and clauses.
- In fine-grained experiments, we find a drop in performance in <polar span, target span, polarity> triplet extraction on our implicit task compared to an explicit sentiment benchmark [15]. Error analysis highlights the need for models that exceed flat lexical inputs due to the lack of strong lexicalization of implicit polar expressions. From this, we provide corpus creation and engineering recommendations regarding implicit sentiment and our dataset.

First, in Section 1.1, we discuss related research on implicit, fine-grained, and economic and financial SA. Next, in Section 2.1, we describe the annotation process, definitions, and properties of the corpus, followed by the IAA study in Section 2.2. Sections 2.3–2.6 describe the experimental methods and data resources used in all validation experiments. Coarse-grained validation experiments are discussed in Section 2.7, followed by the fine-grained triplet modeling approach in Section 2.8. Performance scores are provided in Section 3 and framed in the Discussion (Section 4). Finally, in Section 5, we highlight the major conclusions of our work.

### 1.1. Related Research

This work lies on the crossroads between economics and two recent research strands in the domain of sentiment analysis: implicit sentiment processing and fine-grained sentiment analysis (commonly referred to as aspect-based sentiment analysis (ABSA)). In each subsection, we discuss existing approaches and resources (or lack thereof) for each of these subdomains.

#### 1.1.1. Implicit Sentiment

Explicit statements of subjective intent toward a target entity are called private states [16,17], and are currently the dominant topic of subjectivity analysis research. For instance, the tweet "Most bullish stocks during this dip $GOLD" (from [18]) contains a positive opinion about the company Barrick Gold Corp. (ticker: GOLD) expressed explicitly by the words "most bullish". However, more factual text genres, such as news-wire text or financial reports, often contain implicit expressions of sentiment in the form of objective statements that express a generally desirable or undesirable fact [5]. Van de Kauter et al. [6] found that up to 60% of all sentiment to be implicit in English and Dutch business news. Implicit sentiment analysis targets opinions and attitudes that are not encoded explicitly by lexical expressions but can be inferred through common sense and connotational knowledge. For example the sentence "The computer crashed every day" (from [9]) does not contain any explicit sentiment words, but a negative evaluation of the computer can be inferred from the factual content. Only a small body of research is aimed at implicit sentiment [8,10,11,19,20].

Next, we briefly discuss relevant works on the manual annotation of supervised datasets centered on implicit sentiment tied to factual statements and/or events. Wilson [21] was one of the earliest to annotate implicit sentiment in meetings defining objective polar utterances as "statements or phrases that describe positive or negative factual information about something without conveying a private state". Toprak et al. [9] defined and annotated implicit sentiment for polar facts: sentence descriptions of objectively verifiable facts that imply the quality of an entity or proposition. They annotated these utterances in consumer reviews (e.g., "The camera lasted for many years after warranty." implies a positive attitude toward the camera). In the ISEAR study [22,23], implicit sentiment was labeled as emotions (e.g., anger, joy, etc.) belonging to descriptions of situations, and later made into the EmotiNet structured resource by Balahur et al. [24]. Similarly, the CLIPEval dataset [25] contains sentence descriptions of commonly pleasant and unpleasant activities and life events, and serves as a common benchmark dataset for coarse-grained event-implied sentiment. More fine-grained work can be found in Deng et al. [26], who annotated Good-For/BadFor (gfbf) events that positively or negatively affect entities that strictly fit into

triplets of <agent, gfbg, object>. In later work, Deng and Wiebe [27] detected implicitly expressed opinions by implicature inference over explicit sentiment expressions related to events affecting entities. In Chinese, Huang et al. [28] annotated a corpus of hotel review snippets and clauses for positive and negative polarity. Recently, the SMP2019-ECISA (https://www.biendata.xyz/competition/smpecisa2019/ (accessed on 7 September 2021)) shared task for Chinese implicit sentiment in social media and car product and tourism forum posts based on Liao et al. [7] has spurred new research in implicit polarity modeling. Apart from the annotation of corpora, work on implicit sentiment also led to the creation of connotation lexicons, where words are identified that are superficially objective but have connotational value (e.g., 'delay' is negative), as exemplified by Feng et al. [20] for broad-coverage and Zhang and Liu [19] for consumer reviews.

The related work discussed here only identifies implicit sentiment at coarse-grained levels (the utterance, post, or sentence), often derived from opinionated text genres such as reviews and forum posts. Their is a clear need for a substantial fine-grained resource with token-level polar and target span annotations to enable data-driven approaches for implicit SA.

### 1.1.2. Fine-Grained and Target-Based Sentiment Analysis

Inherent to the definition of implicit sentiment is the inference of a positive or negative attitude change of the reader toward a target entity. Implicit sentiment processing thus contains a component of fine-grained sentiment analysis where the target is of interest. Aspect-based sentiment analysis (ABSA) identifies sentiment of target entities and their aspects [29–33], and has been the dominant line of research in fine-grained opinion-mining. Given a target entity of interest, ABSA methods can identify its properties and the sentiment expressed about those properties.

In ABSA tasks as defined by the SemEval shared tasks [31–33], aspects are pre-defined categories that express properties of a narrowly defined target domain, e.g., the screen, CPU, or battery when focusing on the laptop domain, or the service and food quality in the restaurant domain. This is a good fit for restricted-domain text genres, such as consumer product or service reviews, but does not translate well to the study of more open domains such as business news. Here, restricting targets to certain types would lead to either a proliferation of aspect categories or an omission of potential target annotations. When aspect categories are omitted and targets are processed irrespective of category, this line of research is termed targeted sentiment analysis or target-based sentiment analysis (TBSA). Traditionally, TBSA is narrowly described as target span extraction and sentiment classification (TESC) which involves (a) detecting mentions of target token spans in text (often named opinion target extraction or target/aspect span extraction (TE)) and (b) classifying polarity, a subtask named target/aspect-level/dependent/oriented sentiment (polarity) classification (SC). TE has been widely studied as a separate task in [34–42], as has SC [43–48]. Figure 1 shows the subtasks of fine-grained TBSA that are enabled by our SENTiVENT dataset.

The vast majority of these fine-grained TBSA studies relied on the SemEval ABSA datasets, and occasionally included Mitchell et al. [49]'s open-domain microblog resource or Dong et al. [43]'s microblog resource. The lack of objective genres highlights the need for a manually annotated fine-grained implicit sentiment resource that is not based in opinionated text such as microblogs and reviews.

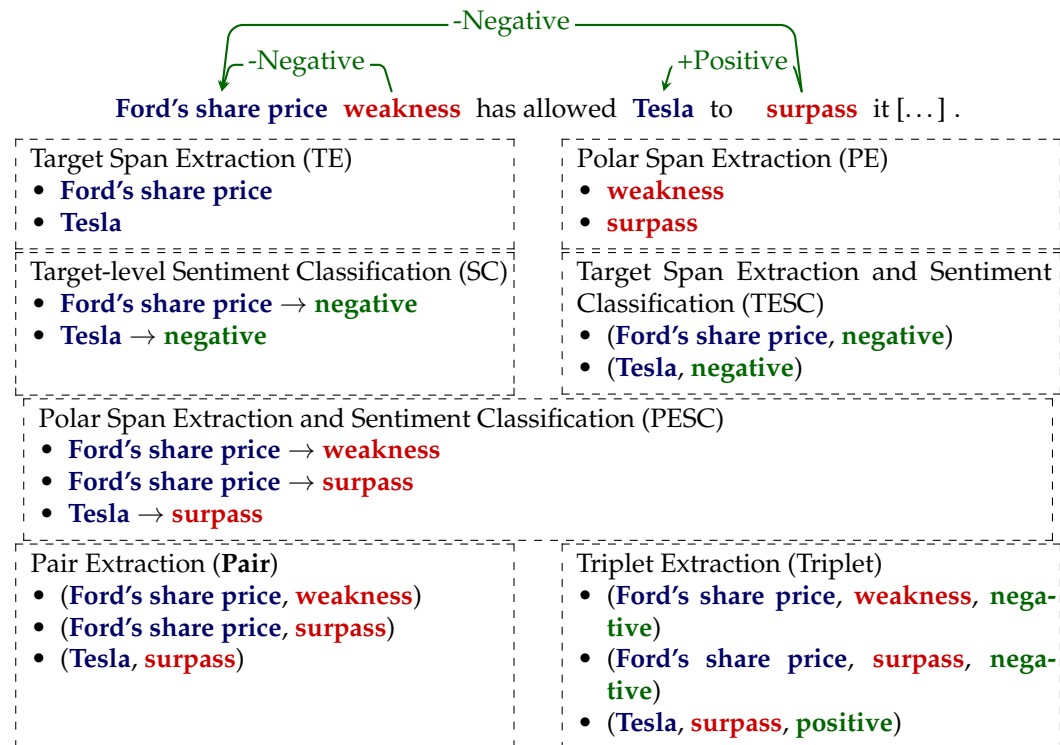

**Figure 1.** Overview of fine-grained target-based sentiment analysis subtasks enabled by the SEN-TiVENT dataset. Overview recreated from Mao et al. [50] but terminology adapted to implicit polar expressions.

Since the organization of the SemEval ABSA shared tasks [31–33] of laptop and restaurant reviews, more fine-grained annotations have been added to these datasets, such as token-level target and aspect spans and separate polarity labels for target-sentiment spans by [15,51], enabling the extraction of <polar span, aspect span, polarity> triplets. Currently, no ABSA or TBSA systems exist that target implicit sentiment.

Most of these studies approached SC with a given gold target span, which limits real-world applicability. Hence, more recently integrated models for both subtasks have been constructed, exploiting associations and commonalities between tasks. Unified models integrating fine-grained subtasks can take advantage of the strong coupling across subtasks: the type of target can provides clues of the sentiment expression and polarity and vice versa. Wang et al. [52], Li et al. [53] applied a sequence tagging approach with a unified tagging scheme to jointly model TESC and show performance competitive with pipelined approaches. He et al. [54] took a multitask learning approach with shared representation learning. Peng et al. [55] were the first to specify <polar span, aspect span, sentiment polarity> triplet extraction in an end-to-end model. They specified a two-stage model in which the first stage comprises sequence labeling unified aspect–sentiment span tags (aspect boundaries with sentiment polarity) and polar spans. The unified labeling approach for aspect–sentiment spans enhanced by learned polar span representations obtaied good results.

### 1.1.3. Financial Sentiment Analysis

Due to the direct economic incentives and value of information acquisition in financial markets, sentiment analysis has attracted a vast amount attention in the field of economics and finance. SA has been applied in various applications such as stock prediction [56–61], financial market analysis [62], impression management of brands or people [2,63–65], macro-economic policy metrics [66,67], and forecasting macro-economic trends and risk [68–70]. Many studies focused on social media text, especially financial microblog StockTwits and tweets [18,71–73]. More objective text types such as periodic

financial reports such as 10-Ks, earnings, and sustainability reports [74–76], as well as news-wire text [58,61,77,78], have also been studied. The lack of explicit sentiment opinions in these genres has led to the study of implicit sentiment in the economic domain. Drury and Almeida [79] identified entity, event, and sentiment phrases in business news by manually creating a rule-based system and assigning them positive or negative polarity. (Unlike the event annotations in SENTiVENT (ours), these are not manually labeled and syntactically limited to verb phrases. The concept of events here is agentive actions by economic actors and is heavily restricted into five action categories. The sentiment words were identified by the expansion of a seed lexicon and matching; hence, they are derived from lexically explicit sentiment.) However, by combining the events and polarity with this rule-based method, they were one of the first to study implicit sentiment in economic news. Similarly, Musat and Trausan-Matu [10] interpret implicit sentiment that emerges through the co-occurrence of economic indicators (e.g., unemployment) and future state modifiers (i.e., spans indicating the growth or decrease in the economic indicator to which they are referring).

Malo et al. [80] annotated around 5000 sentences of English news and press releases of Helsinki Stock Exchange listed companies for polarity. The resulting Financial Phrasebank is an often-relied-on benchmark dataset for coarse-grained financial SA. Chen et al. [81] compared writer-labeled vs. market-expert-labeled microblogs (labeling your tweet bullish/bearish is a feature in StockTwits) in order to investigate discrepancies. Regarding fine-grained resources, Cortis et al. [82] created the SemEval-2017 Task 5 dataset for fine-grained economic SA consisting of microblogs (2510 messages) and news headlines and statements (1647 headlines). Span annotations for the opinion words were made; targets were identified at the document level and assigned to polar spans with sentiment polarity scores. This resource thus enables target-based economic polarity classification but it lacks fine-grained target span annotations. Maia et al. [83] presented the FiQA'18 Task 1 dataset for economic ABSA, containing a set of 529 headlines and 774 microblogs, including target annotations, sentiment scores, and aspect categories.

Many of the existing approaches to (financial) SA remain coarse-grained: they detect the mood of a certain document by taking into account all expressions of sentiment, regardless of the target. Few fine-grained financial resources exist [82,83] they are limited in scope and size, and/or are pre-filtered to ease annotation: FiQA'18 only includes sentence instances in which target companies are explicitly named (viz. realized as nominal phrases, e.g., 'Berkshire Hathaway' or 'Tesco'); the SemEval-2017 Task 5 dataset does not contain target span annotations. All previous work has either been conducted in opinionated and/or subjective text genres such as microblogs or on news headlines instead of full article text. This work fills the need for a manually labeled implicit (and explicit) economic sentiment dataset, while enabling fine-grained SA in full news articles.

Importantly, the concept of sentiment in finance is different than in NLP: where market or investor sentiment is defined as "the expectations of market participants relative to the norm", a bullish/bearish investor expects returns to be above/below average, whatever average may be [84]. Additionally, sentiment definitions in economics and finance often depend on a specific application. Consumer confidence is the general consumer expectations about the future state of the economy. In behavioral finance, Long et al. [85] showed that investors are subject to sentiment that is not strictly tied to fundamentals. Baker and Wurgler [86] defined investor sentiment as "a belief in future cash flows and investment risks that is not justified by the facts at hand" and provided several proxies to measure sentiment such as surveys, retail investor trades, mutual fund flows, dividend premiums, trading volume, option-implied volatility, etc. Kearney and Liu [87] proposed two general types of sentiment definition: (a) investor sentiment, which constitutes the subjective judgements and behavioral characteristics of investors; and (b) textual sentiment, which entails text-based expressions of positivity/negativity, and can also include investor sentiment and adds "the more objective reflection of conditions within the firms, institutions, and markets". Our operationalization of investor sentiment as event-implied polar expressions

fits this latter definition as we annotate the sentiment polarity of factual real-word events. We opt for the term textual investor sentiment to highlight the presumption of an imagined investor affected by the expressions of both explicit and implicit opinion in the text. This definition encompasses both implicitness and targets: it depends on the inferred attitude of an assumed reader-investor toward a target entity.

### 1.2. Previous Work

Our work is a continuation of the English and Dutch SentiFM business news corpus [6,88], which contains token-level annotations of implicit and explicit sentiments, targets, sources, source-linkers, modifiers, and cause spans. On this corpus, we experimented with coarse-grained event detection [89]. However, SentiFM events are not directly related to sentiment, and experiments remained coarse-grained.

The SENTiVENT-Event dataset [14,90] of ACE/ERE-like event schemata [91,92] contains a typology of economic business with prototypical argument roles. We enhanced the SENTiVENT-Event dataset by (a) annotating explicit and implicit sentiment–target pairs, and (b) adding sentiment polarity labels to event triggers (i.e., the word span denoting an event). This results in a fine-grained, token-level dataset with sentiment–target annotations enabling fine-grained extraction of facts (as events) and subjectivity (as implicit sentiment) to enable target-based implicit SA.

## 2. Materials and Methods

We experimented with multiple sentiment analysis approaches for the detection of implicit company-dependent sentiment in economic news text. Lacking resources for this task, we applied a fine-grained annotation scheme covering polar facts and sentiment expressions. Section 2.1 describes definitions of labeled units, dataset properties, and the annotation process with a focus on validation through an inter-annotator agreement (IAA) study. Then, Section 2.3 specifies the experimental setup for the coarse- and fine-grained experiments, model architectures, model selection, and validation testing.

### 2.1. Data Collection and Annotation

In this section, we present the definitions, annotation process, and corpus properties of the English SENTiVENT financial news dataset for implicit fine-grained sentiment processing. The sentiment annotations are made in the SENTiVENT-Event English corpus of fine-grained event annotations described in previous work [14,90]. The dataset consists of a random crawl of full online news articles mentioning specific companies between the period of June 2016 and May 2017. The companies were randomly chosen from the S&P500 index, but sectorial diversification was ensured to avoid topical specialization. The applied definition of implicit sentiment is the following:

**Textual investor sentiment**: The definition of annotated subjectivity is tailored to the economic and financial domain, with the goal of enabling micro-economic information extraction and market analysis. Investor sentiment is defined as textual expressions that encode (explicitly) or affect (implicitly) investor attitude and/or opinion toward a target entity in the market. An investor is someone looking to assign capital to a company, asset, or other entity with the expectation of profit in the future. Investor sentiment can be defined as the opinions that an investor holds toward a potential investee. In financial terminology, positive economic expectations are called bullish and negative expectations, bearish. We annotate positive (bullish), neutral, and negative (bearish) polarity.

The polar expression annotations consist of the following parts exemplified in Figure 2:

- Polar span: the token span expressing the implicit or explicit investor sentiment.
- Target span: the continuous token span denoting a sentiment target entity; the company, person, organization toward which the sentiment is directed.
- Polarity:
    - Positive investor expectations come from events that have a desirable effect on a business entity's characteristics (e.g., its financial metrics, growth, and position in

the market) or the larger surrounding economic situation (e.g., macro-economic factors, policy changes, and market fluctuations). Examples of positive polarity are increases in growth of sales, revenue, profit, cash flow, or other financial metrics; strategic investments; cut expenses; well-reviewed products; effective marketing efforts, growing stock price or increased price targets; upgraded ratings; optimistic analyst expectations; etc.

– Negative investor sentiment entails the opposite expectation that a loss will occur and the invested funds will not generate an acceptable return. Generally, negative expectations come from events that have an undesirable effect on some attribute or the surrounding situation of a business entity. Examples include inhibition of growth or decline in sales, revenue, profit, cash flow, and other financial metrics; or scandals, losses, legal issues, increased expenses, lowering stock price or price targets, downgrading of ratings, employment issues, negative analyst expectations, etc.

– Neutral investor sentiment is for an expression with no clear positive or negative polarity. This happens when (a) the polar expression expresses ambivalence on the part of the author, or (b) it is unclear/ambiguous how the polar expression could affect a potential investor's attitude.

- Two factuality attributes [93]:

  – Non-certain modality: the sentiment expression is presented as being anything other than certain (i.e., certain epistemic modality).

  – Negation: the sentiment expression is in the scope of negation, i.e., the author indicates the sentiment/polar event is not the case.

Our sentiment annotations are of two types:

(a) Directly-annotated polar expressions consisting of polar spans with target span(s) and sentiment polarity annotations, as shown in Figure 2a,b.

(b) Event-induced polar expressions: sentiment polarity labels added on top of the event trigger annotations to encode implicit sentiment, i.e., connotational common-sense sentiment expressed by these events, as shown in Figure 2c.

The first type of annotations (a) is labeled everywhere in the article text where a preexisting event trigger is absent or does not adequately describe the sentiment (except when the trigger is longer or shorter than a polar span). Polar spans can be tagged with multiple or no targets and cross-sentence polar–target span relations are allowed. Positive, neutral, or negative polarity is target-dependent; occasionally, one polar span can be negative for one target and positive for another.

The preexisting event annotations of type (b) stem from the previously mentioned SENTiVENT-Event corpus. Events denote real-world occurrences drawn from a representative typology to encode prototypical event types and their argument roles. These event schemata allow for fine-grained annotation of commonly reported changes, actions, and state of affairs in company-specific news and the argument entities that take on typical roles within them. An event annotation consists of:

- Trigger: the shortest span of words expressing the event.
- Arguments: spans of words denoting predefined prototypical roles in the event of a certain type (48 different roles in the typology).
- Type and attribute labels: the event type, subtype (18 types and 42 subtypes in the typology), and several other attributes (modality, negation, and realize).

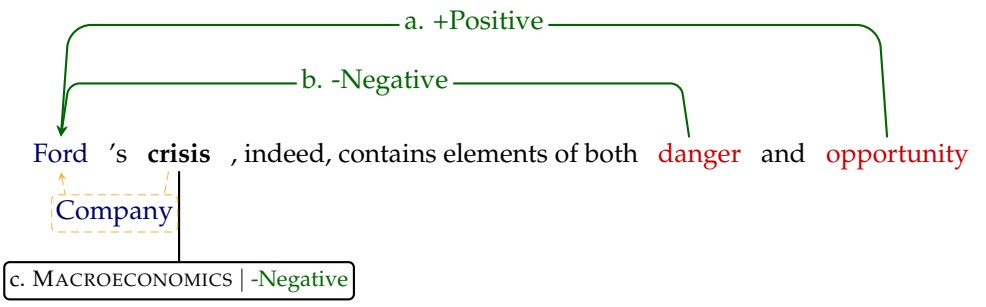

a. Positive implict sentiment targeted to "*Ford*" expressed by '*opportunity*'.

| Directly-annotated polar expression | |
|---|---|
| Polarity | +Positive |
| Target Span | "*Ford*" |
| Polar Span | "*opportunity*" |
| Not negated, not non-certain | |

b. Negative implict sentiment targeted to "*Ford*" expressed by "*danger*".

| Directly-annotated polar expression | |
|---|---|
| Polarity | -Negative |
| Target Span | "*Ford*" |
| Polar Span | "*danger*" |
| Not negated, not non-certain | |

c. Event-induced negative sentiment targeted to argument "Ford".

| Event | |
|---|---|
| Trigger | "*crisis*" |
| Type | Macroeconomics |
| Arguments | |
| Company | "*Ford*" |
| Not negated, not non-certain | |

⇒

| Event-induced polar expression | |
|---|---|
| Polarity | -Negative |
| Target Span | "*Ford*" |
| Polar span | "*crisis*" |
| Not negated, not non-certain | |

**Figure 2.** Examples of two directly annotated implicit target sentiment expression annotations (**a**,**b**) and one event-induced sentiment (**c**).

Figure 2 contains an event schema of the Macroeconomics type as expressed by the event trigger "crisis" and it has one argument "Ford" of the role "Company". Figure 3 shows two events: one of type Revenue expressed by trigger "load factor increase" with the argument role Company pointing to "American Airlines", and Time to "May", and another of type SecurityValue with the trigger "shares gain" and to "American Airlines" as the Security argument.

To obtain polar expressions from event schemata, we annotated the polarity labels of the event and mapped the argument spans to targets. The event trigger span becomes the polar span expressing the sentiment. The argument-to-target is achieved by filtering argument roles in the event typology: arguments that are generally non-agentive or non-central are left out and cannot be targets. In Figure 3, it is clear the Time argument cannot be the target of the positive effect of an increase in revenue as the time of an event is incidental. On the other hand, the Company role that experiences the event is the primary benefactor, so the positive event maps the Company role to the target. Examples of removed argument roles are those referring to amounts of capital (Dividend.YieldRatio, Expense.Amount, and Employment.Compensation), or incidental descriptors such as Time and Location, results of events (Deal.Goal,Legal_Conviction.Sentence), or non-central entities on which the event polarity cannot be reflected (Rating.Analyst). Generally, central and agentive roles that are kept as targets are filled by companies, people, or organizations (CSR/Brand.Company, Employment.Employer). This mapping takes advantage of event-level implicit polarity, reflecting and precipitating on the pre-defined argument roles. In exceptional cases, this assumption does not hold and annotators are directed to make separate directly annotated polar expressions for each target-specific polarity.

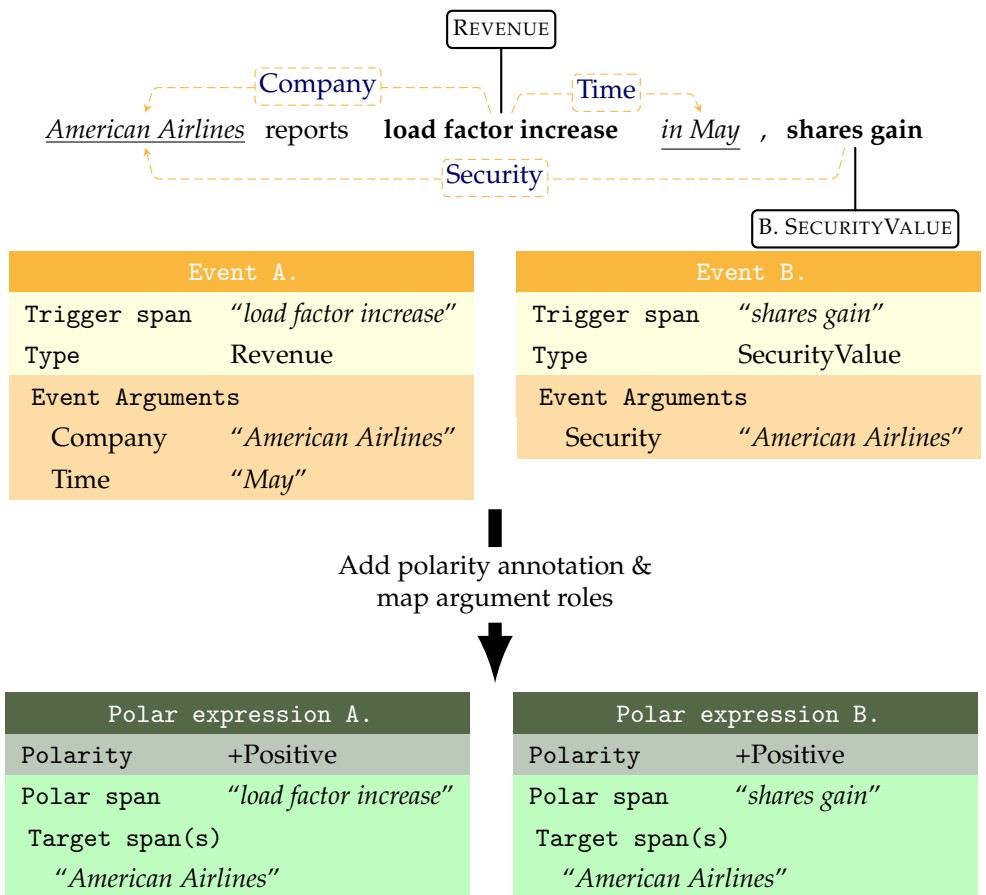

**Figure 3.** Example of event-induced polar expressions obtained by adding implicit polarity annotation and mapping argument roles to targets.

Table 1 shows the properties and annotation counts of our corpus. The corpus is larger than the currently largest fine-grained TBSA corpora from Wu et al. [15] (8937 polar spans, 9337 target spans, and 10,390 triplets), Mitchell et al. [49] (English: 3288 tweet-target pairs, Spanish: 6658), Dong et al. [43] (6940 tweets, unknown number of targets). Note that the count of triplets, i.e., unique sets of <polar span, target span, polarity>, is larger than the polar span count and target count because one polar span can be related to $n$ multiple targets, viz. triplets are a product result of <polar span, (target span, polarity)$_i \in \{i, \dots, n\}$>).

Figure 4 shows the frequency distribution of polarity labels by their annotation source. We can observe that positive/bullish sentiment is more common than negative/bearish sentiment. A larger amount of bullish sentiment was also observed by Chen et al. [81] in their Stocktwits dataset. In our dataset, the skew is in part due the established value and stability of the selected companies, which all stem from the S&P500 index during an up-trending period. This leads to a lower likelihood of unpleasantly surprising events. The high frequency of neutral polarity in events is due to the larger likelihood of ambiguous or weak sentiment of event descriptions.

We define a standard split for reference use of the dataset (i.e., hold-in training, hold-in development evaluation, and hold-out evaluation set, cf. Table 1). The test set corresponds to the 30 documents of the IAA study consisting of the combined and corrected annotations by three different annotators. This provides a high-quality, robust dataset for evaluation of the best model.

**Table 1.** Properties and annotation counts for the total corpus and training, development, and holdout test sets.

| Corpus Property | Total | Train | Dev | Test |
|---|---|---|---|---|
| Polar spans | 9909 | 7587 | 1010 | 1312 |
|    Directly annotated | 3669 | 3531 | 387 | 328 |
|    Event-induced | 6240 | 4633 | 623 | 984 |
| Target spans | 10,889 | 8128 | 1139 | 1622 |
|    Directly annotated | 2920 | 2306 | 313 | 301 |
|    From event arguments | 7969 | 5822 | 826 | 1321 |
| Triplets | 12,398 | 9353 | 1309 | 1736 |
|    Directly annotated | 4429 | 3531 | 483 | 415 |
|    Event-induced | 7969 | 5822 | 826 | 1321 |
| Documents (full articles) | 288 | 228 | 30 | 30 |
| Sentences | 6883 | 5475 | 681 | 727 |
| Tokens | 170,398 | 135,317 | 17,955 | 17,126 |

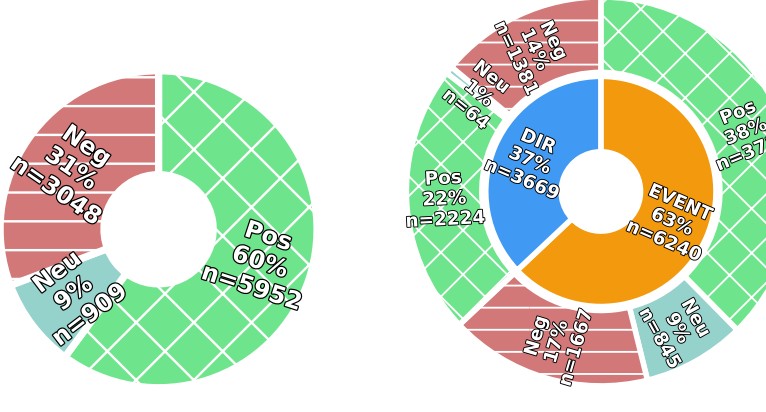

(**a**) Polarity frequency        (**b**) Polarity frequency by source

**Figure 4.** (**a**) Total frequency of positive (Pos), neutral (Neu), negative (Neg) polarity, and (**b**) polarity subcategorized by event-induced (EVENT) or directly-annotated (DIR) polar expressions.

### 2.2. Inter-Annotator Agreement Study

As our task demands a large degree of interpretation (implicit sentiment requires inference by its definition, the polarity has the potential to be ambiguous), we conducted an inter-annotator agreement study to determine whether our annotation guidelines were properly defined and resulted in consistent and reproducible annotations. A subset of 30 documents discussing ten companies were included corresponding to ±10% of the full dataset. Each document was fully annotated by three annotators. We used chance-corrected agreement measures Fleiss' Kappa [94] $\kappa$, Krippendorf's alpha [95] $\alpha$, and Gwet's AC1 [96] on aligned span annotations and used the R package irrCACC (https://CRAN.R-project.org/package=irrCAC) (accessed on 7 September 2021) to compute these metrics. Fleiss' $\kappa$ [94] generalizes Scott's Pi to any number of annotators. Krippendorff [95] argued for the use of $\alpha$ over other measures such as $\kappa$ because of its independence from the number of assessors and its robustness against imperfect data. AC1 was introduced by Gwet [96] to provide a more robust measure of interrater reliability than Fleiss' $\kappa$. AC1 can overcome $\kappa$'s sensitivity to trait prevalence and rater's classification probabilities (i.e., marginal probabilities) [97]. (We did not use weighting or custom label distances when assessing disagreement, even though, for sentiment polarity, mislabeling neutral as positive or negative is much less of an error than labeling positive as negative. Discounting this type of disagreement would have provided higher scores for sentiment polarity as confusion of neutral with positive is by far most common).

In line with Lee and Sun [98], we computed several agreement metrics by using relaxed span matching for directly annotated polar expressions using span overlap. This allowed us

to determine agreement on the labels of annotation units where the exact span overlap was not of interest (cf. overlapping span matching in Appendix A). The substantial agreement of our overlap-matched spans shows that the general area of span annotations is mostly agreed on despite the unmatched boundaries. Other work on ABSA resources performed agreement scoring at the sentence level [33,99] or clause level [100], thus circumventing the issue of matching token-level annotations, but they also lost granularity.

Table 2 shows that our annotations obtained adequate agreement scores on all categories. We benchmarked these coefficients using the cumulative membership probabilities (As recommended by Gwet [101], we set the benchmark membership probability cut-off point at 95%) within the agreement ranges as set out by Landis and Koch [102]: $\kappa \leq 0$ = poor agreement, $0 \leq \kappa \leq 0.2$ = slight agreement, $0.2 \leq \kappa < 0.4$ = fair agreement, $0.4 \leq \kappa < 0.6$ = moderate agreement, $0.6 \leq \kappa < 0.8$ = substantial agreement, and $0.8 \leq \kappa \leq 1$ = (almost) perfect agreement. For polarity, agreement obtained a Fleiss' $\kappa$ of 0.78, an AC1 of 0.85, and an $\alpha$ of 0.78 signifying substantial agreement within the 0.6–0.8 range. On negation, $\kappa$ is 0.74, $\alpha$ is 0.81, and AC1 is 0.95. For $\kappa$ and $\alpha$, this signifies substantial agreement, while for AC1, this falls in the almost perfect agreement. On non-certainty, $\kappa$ is 0.60, AC1 0.82, and alpha is 0.62. For $\kappa$ and $\alpha$, this signifies moderate agreement, while for AC1, this falls into substantial agreement. Non-certainty has lower agreement than the other categories, since annotators experienced difficulty in determining when sentiment was in the scope of modal non-certainty. While processing non-certainty is important for downstream tasks in which the factuality of sentiment and events influences decision making, it was not the focus of this work.

**Table 2.** Directly-annotated polar expression agreement scores on polarity, negation, and non-certain labels.

| | Metric | Coeff | SE | 95% C.I. | *p*-Value |
|---|---|---|---|---|---|
| Polarity | | | | | |
| | Fleiss' $\kappa$ | 0.778 | 0.050 | (0.68, 0.876) | <0.0001 |
| | Krippendorff's $\alpha$ | 0.782 | 0.035 | (0.714, 0.851) | <0.0001 |
| | Gwet's AC1 | 0.849 | 0.045 | (0.76, 0.938) | <0.0001 |
| Negation | | | | | |
| | Fleiss' $\kappa$ | 0.736 | 0.081 | (0.576, 0.896) | <0.0001 |
| | Krippendorff's $\alpha$ | 0.809 | 0.051 | (0.709, 0.909) | <0.0001 |
| | Gwet's AC1 | 0.947 | 0.044 | (0.861, 1) | <0.0001 |
| Non-certain | | | | | |
| | Fleiss' $\kappa$ | 0.601 | 0.067 | (0.469, 0.733) | <0.0001 |
| | Krippendorff's $\alpha$ | 0.619 | 0.056 | (0.51, 0.728) | <0.0001 |
| | Gwet's AC1 | 0.821 | 0.046 | (0.73, 0.913) | <0.0001 |

For pre-existing events, annotators had to assign positive, neutral, or negative sentiment polarity. Table 3 shows the agreement analysis for polarity labels of events. Fleiss' $\kappa$ and Krippendorf's $\alpha$ are both 0.65 and Gwet's AC1 is 0.72. Following [102], the agreement scores range between moderate ($\kappa$ and $\alpha$) and substantial (AC1). The sentiment polarity annotations on existing events thus show lower agreement than on the newly annotated sentiment spans. This is readily explained by the increased difficulty in the interpretation of the polarity invoked by events. Annotators more easily recognized sentiment span annotations as they were more clearly positively or negatively polar. As also shown in Figure 4, neutral polarity is much more common for the events than in directly annotated polar expressions (9% vs. 1%).

Overall, we thus conclude that our annotations are adequate for use in polarity and negation tasks showing substantial agreement; ideally, the annotation guidelines for non-certainty should be revised with expanded rules for annotating this category.

**Table 3.** Polarity agreement on events.

| | Metric | Coeff | SE | 95% C.I. | *p*-Value |
|---|---|---|---|---|---|
| **Polarity** | | | | | |
| | Fleiss' $\kappa$ | 0.648 | 0.017 | (0.614, 0.682) | <0.0001 |
| | Krippendorff's $\alpha$ | 0.648 | 0.017 | (0.614, 0.682) | <0.0001 |
| | Gwet's AC1 | 0.724 | 0.015 | (0.695, 0.754) | <0.0001 |

*2.3. Experimental Method*

We present three series of experiments from coarse- to fine-grained in order to validate the created dataset and the feasibility of fine-grained implicit sentiment detection in economic news: implicit polarity classification on (1) gold polar expressions, (2) clauses, and (3) end-to-end extraction of <polar span, target span, polarity> triplets.

First, in Section 2.7, we test multiple large-scale pre-trained language models (PLMs) in a coarse-grained classification set-up with only gold instances of polar expressions (i.e., spans containing positive, neutral, or negative sentiment polarity). The task here is to classify implicit positive, neutral, and negative polarity given the known token span of a polar expression and its targets. This is a preliminary series of experiments aimed to test the viability of implicit polarity classification: it is missing the required detection of polar expressions in a real-world setting. This simplified task allowed us to select the most viable pre-trained language model encoders for the fine-grained and increasingly difficult tasks. Subsequently, we applied coarse-grained, clause-based, implicit polarity classification, which includes a None class. Our original token-level annotations were transformed into sentential subclauses and assigned an implicit polarity label. Next, we applied coarse-grained, clause-level, implicit sentiment analysis where each sentence was split into its clauses and polarity was classified. This presents a more realistic setting since clause instances without implicit polarity are included. Clauses are also a good fit for implicit sentiment classification as they constitute fully realized semantic units that correspond well to polar expressions. Finally, in Section 2.8, we apply end-to-end triplet extraction to test the viability of state-of-the-art approaches in explicit sentiment modeling on implicit data. This represents the most fine-grained and complete task taking full advantage of the fine granularity of our sentiment annotations.

All three series of experiments relied on the same transformer-based PLM encoders, which are discussed in Section 2.5. For the coarse-grained experiments, we tested adding external subjectivity lexicons (both general and in-domain) as features in classification. These are discussed in more detail in Section 2.6. For the coarse-grained experiments, we applied the same model selection approach and evaluation approach; only the evaluation approach for the fine-grained experiments differed.

Model selection: We ran hyperparameter optimization maximizing the macro-$F_1$ score using a Bayesian search with hyperband stopping [103] for 128 runs. Optimized parameters included learning rate, batch size, and lexicon feature set. We used different hyperparameter search spaces for base- or large-size variants based on explorative experiments (cf. Appendix D for the full search space). We also enabled early stopping within runs, evaluating inter-epoch macro-$F_1$ on the devset, selecting the best epoch of each run. This eliminated the need for including the number of epochs as a hyperparameter to optimize. The highest scoring model on devset macro-$F_1$ in the search was selected as the winner.

Model evaluation: For coarse-grained experiments, first, each winning development architecture was evaluated on the holdout test set to check for overfitting. Second, we performed a significance test on the test set predictions of each: first, we performed Cochran's Q test across all model predictions. Cochran's Q tests the hypothesis that there is no difference between the classification accuracy across all architectures. If this hypothesis was rejected, we performed a pairwise McNemar's significance test to compare the best-scoring model across architectures to all others, in line with the recommendations of Dror et al. [104]. For fine-grained experiments, we selected the best model hyperparameters by

the $F_1$-score on triplet extraction. In line with Wu et al. [15], we re-trained and hold-out tested five times with different random seeds. The final holdout test score is the average of these five random initializations.

### 2.4. Pre-Processing

Prior to experimentation, we performed linguistic and annotation pre-processing. Sentence-splitting and tokenization were performed before annotation using StanfordCoreNLP [105] to allow token-level annotations. For use with sentiment lexicons, we also applied part-of-speech tagging, stemming, and lemmatization using Spacy.

The event-to-polar mapping approach, as described in Section 2.1, was applied to obtain a dataset of polar expressions. We removed pronominal realizations (anaphoric "it", "that", and "they") of polar expressions and targets if there was no non-pronominal referent annotation present as part of the event coreference annotations. We also removed cross-sentence target relations: if a polar span was in a different sentence than its target span, we removed that relation.

### 2.5. Pre-Trained Models

We performed all experiments using the fine-tuning of PLM transformers BERT [106], RoBERTa [107], and DeBERTa [108]. Additionally, we used domain-specific versions of BERT that were pre-trained and/or fine-tuned on economic news or other financial corpora.

BERT is an attention-based autoencoding sequence-to-sequence model using two unsupervised task objectives: The first task is word masking or masked language model (MLM), where the model guesses which word is masked in its position in the text. The second task is next sentence prediction (NSP) performed by predicting if two sentences are subsequent in the corpus, or if they are randomly sampled from the corpus.

The RoBERTa model [107] is an improvement over BERT by dropping next sentence prediction and using only the MLM task on multiple sentences instead of single sentences. The authors argued that while NSP was intended to learn inter-sentence coherence, it actually learned topic similarity because of the random sampling of sentences in negative instances.

DeBERTa [109] improves upon the performance of BERT and RoBERTa by using a disentangled attention mechanism and an enhanced mask decoder. The disentangled attention mechanism represents each word with a position and content vector and using disentangled attention matrices to compute attention-weights-based contents and relative positions of word pairs (more specifically, attention is computed as the sum of four attention scores of word pairs using separate matrices on contents and position such as content-to-content, content-to-position, position-to-content, and position-to-position). The enhanced mask decoder introduces absolute position in the masked-token prediction objective right before the SoftMax-decoding layer (unlike BERT, where absolute positions are added at the input layer).

We also tested two in-domain fine-tuned models that were trained on financial reports and economic news text data. Araci [110] (henceforth FinBERT$_{TRC2+FP}$) applied further pretraining of a general domain BERT model on the TRC2-financial corpus consisting of business news articles. Subsequently, the model was fine-tuned on the Financial Phrase-bank, classifying sentence-level positive, neutral, and negative polarity in financial news. The model is a good fit for our purposes as it was further pretrained on in-domain data and fine-tuned on a task analogous to ours. Additionally, we tested Yang et al. [111] (henceforth FinBERT$_{FinVocab}$), which is a BERT model trained from scratch on corporate reports, earning call transcripts, and analyst reports. This model uses its own in-domain SentencePiece vocabulary and obtains good results when fine-tuned on the Financial Phrasebank [80], FiQA [83], and AnalystTone [112] tasks.

For all transformer models, we used the available base-size with 12 layers, 768-dimensional hidden representations, 12 transformer heads, and 125 million trainable parameters. For BERT and RoBERTa, we also included the large size of 24 layers, 1024-

dimensional hidden representations, 16 transformer heads, and 355 million parameters. We used model variants that maintain casing vocabulary (cased), except for FinBERT-FinVocab, for which we used the lowercasing variant (uncased) as this obtained best performance across all sentiment tasks in the original work [111]. The references for the pretrained model weights as well as more details on the classification head can be found in Appendices B and C.

### 2.6. Subjectivity Lexicon Features

In order to leverage existing SA resources, we included several general domain and economic- and financial-domain lexicons. Table 4 lists the lexicons used, their domain, and briefly describes their creation, categories, word-matching approach, and how we used them. We experimented with various different methods of computing and normalizing the final polarity score of a word sequence as a hyperparameter in model selection: the lexicons matched one or more words to categories, often including polarity such as positive or negative but also other categories of emotive or psycho-cognitive nature. We experimented with generating feature vectors with match counts for all categories, and combinations of methods of computing a final polarity score for the whole text sequence in the instance. Different lexicon feature-sets were generated, including direct wordlist match counts (e.g., raw, length, or match-normalized counts of negative, positive, neutral, or other specific sublists of lexicons such as Money and Anxiety in LIWC) and several methods of sequence-level polarity scoring (i.e., nominal match counts, token-length normalized, total match-normalized), were implemented and optimized in model selection. In coarse-grained experiments (Section 2.7), we compared using no lexicons, only the domain-specific economic lexicons, and combinations of general and economic lexicons. These feature sets were grouped into three settings: polarity includes only the sequence polarity estimations;polarity + polarwordlists adds subwordlist counts for categories that denote positive, neutral, or negative polarity (if present in the lexicons); and all adds wordlist counts of non-direct polar categories. The selection of feature sets within the combination of lexicons was included as a hyperparameter in the hyperparameter optimization search as we did not consider the variations in lexicon feature sets a major architectural feature.

**Table 4.** Overview of subjectivity lexicons used in coarse-grained implicit sentiment experiments.

| Lexicon | Description | Domain |
|---|---|---|
| Henry [113] | Early manually made lexicon for positivity and negativity in tone of earnings press releases. Token matching. | Economic |
| NTUSD-Fin v1 [114] | Token lexicon with market sentiment scores (bullish-bearish) induced from annotated social media microblogs StockTwits. Token matching. | Economic |
| Loughran-McDonald Master v2018 (LM) [115–117] | Token lexicon of exclusively economic terms initially developed manually on financial reports but expanded in later studies. Includes subwordlist positive, negative, uncertainty, litigious, constraining, interesting, and modal; we excluded the general domain HIV4 subwordlist and the very small superfluous sublist from our experiments. Token matching. | Economic |
| SentiEcon v1 [118] | Single and multiword expression lexicon derived from business news through iterative refinement. Includes positive, negative, neutral polarity tags; for our experiments, we did not consider polarity intensity values. Multiword expression matching. | Economic |
| Macquarie Semantic Orientation Lexicon v0.1 (MSOL) [119] | High-coverage automatically bootstrapped general-domain lexicon for positive and negative polarity. Lemma matching. | General |
| SentiWordNet3 [120] | Lexicon of automatically annotated WordNet synsets for positivity, negativity, neutrality. We used geometric weighting across senses to derive polarity scores in line with the recommendations in [121]. Multiword expression matching. | General |
| LIWC [122] | Linguistic Inquiry and Word Count (LIWC) contains several emotion, cognitive, and social categories and focuses on the psychometric aspects of subjectivity. We omitted several of the linguistic categories, but kept subjectivity and business-related categories. We also grouped positive and negative emotion/psychosocial categories for the purpose of polarity score calculation. Stem matching. | General |

The lexicon match and polarity score vector was appended to the embedded representation produced by the transformer and input into the classification head. This allowed us to compare and use existing sentiment resources for implicit polarity classification.

*2.7. Coarse-Grained Experiments: Implicit Polarity Classification of Gold Polar Expressions and Clauses*

Our dataset contains token-level annotations that enable fine-grained implicit sentiment analysis. However, given the increased difficulty of implicit sentiment analysis vs. regular explicit tasks, we first checked the feasibility of implicit polarity classification at a more coarse-grained level in two experiments:

(a) Implicit polarity detection for gold polar expressions: classify implicit polarity {Pos, Neu, Neg} of known polar expressions, i.e., the known sequence of tokens denoting the implicit sentiment expression and its target(s). This does not require the detection of polar spans or target spans as they are given as input (e.g., Figure 5a).

(b) Clause-based implicit polarity detection: detect the implicit polarity {None, Pos, Neu, Neg} of sub-sentence clauses, where None indicates the absence of polarity (e.g., Figure 5b).

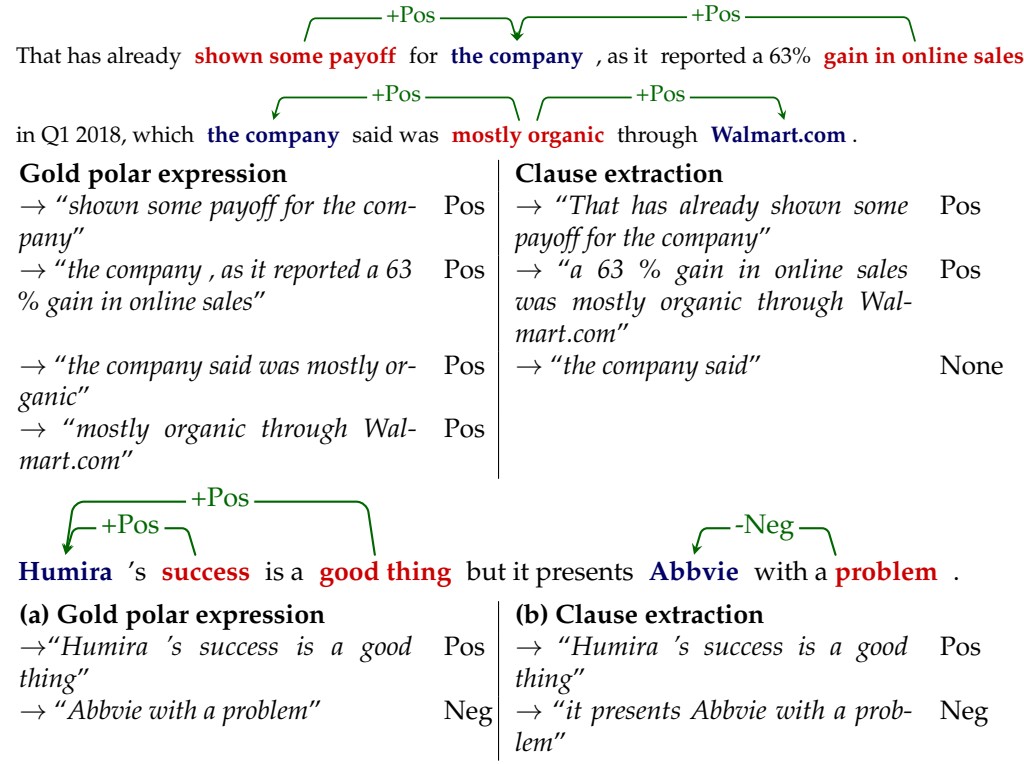

**Figure 5.** Fine-grained annotations to coarse-grained text inputs with label for (**a**) gold polar expression (i.e., the labeled polar span and target span made continuous) and (**b**) clause-based experiments.

For (b), we split the corpus sentences into sub-clauses using the OpenIE 5.1 (https://github.com/dair-iitd/OpenIE-standalone (accessed on 7 September 2021)) clause extractor [123,124] with a rule-based dependency-parsing fallback when OpenIE fails. When multiple polar expressions with different polarities are present in the clause, we assigned the label by majority vote, with positive and negative taking precedence over neutral in the case of a draw. OpenIE clauses are a good fit as they are a triplet representation of <subject, relation, argument(s)> where each part is a phrase. The OpenIE clause is conceptually similar to the concept of polar expressions, while providing a more fine-grained content

analysis than classifying full sentences. A major difference from polar expressions and OpenIE clause triplets is that a subject–argument distinction does not exist for targets, and in OpenIE, the relational phrase is very often a verb phrase, whereas this is not the case for polar expressions, which can be any token span. However, this approach introduces the detection of relevant polar expressions as clauses, which is useful in applications for retrieving positive, neutral, and negative sub-sentence sequences that correspond to a singular description of a state of affairs involving arguments.

For both these tasks, the token spans were encoded by the transformer model with sequence classification head described in Section 2.5 and, if applicable, the lexicon features were computed. The transformer model was fine-tuned through all layers on the implicit polarity classification task.

### 2.8. Fine-Grained Experiments: Implicit Triplets

Since our annotations are fine-grained at the token level, an end-to-end extraction task for <polar span, target span, polarity> is a good fit. This task has recently been enabled by a benchmark dataset released by Fan et al. [51], which adds annotated target–opinion pairs based on SemEval ABSA challenges [31–33]. These datasets are typical examples of explicit sentiment in reviews of restaurants and laptops. Wu et al. [15] aligned these annotations with the corresponding sentiment polarity to obtain token-level triplet annotations. We compared the performance of a state-of-the-art model in explicit triplet extraction to our implicit dataset to test the applicability and transferability of the currently best available approach. Instead of evaluating these sub-datasets separately, we joined all instances into one large dataset (henceforth, Explicit Wu et al. [15]) to be more comparable in size and domain-diversity to our dataset; however, the diversity in text genre and domain (i.e., consumer reviews) remained highly limited compared to SENTiVENT economic news.

Alongside the benchmark Explicit dataset, Wu et al. [15] presented a novel grid-tagging scheme (GTS) architecture for TBSA that operationalizes triplet extraction as a unified tagging task across a grid representation. In contrast with more common pipelined approaches that suffer from error propagation, all word-pair relations are tagged and all opinion pairs simultaneously decoded. Figure 6 shows all word-pair tags in a simplified upper triangular grid for an example sentence. GTS involves two steps: (1) a unified tagging task and (2) a decoding step to robustly decode predicted word-pair tags along spans where polar spans and target/aspect spans are linked if a word pair containing a polarity tag is present.

(1) The unified grid-tagging scheme uses the tagset $\{T, P, \text{POS}, \text{NEU}, \text{NEG}, \varnothing\}$ to denote the relation of any word pairs in a sentence. POS, NEU, NEG correspond to positive, neutral, and negative sentiment, respectively, expressed in the opinion triplet of word pair $(w_i, w_k)$. Table 5 shows the meaning of the tags and Figure 6 shows the result of tagging a sentence in the form of an upper triangular grid.

(2) The decoding stage involves relaxing constraints on the predicted tags in a sentence. Strictly matching target and polar span relations of the word-pairs would suffer from low recall due to the majority of $\varnothing$ tags. First, the predicted tags on the main diagonal are used to recognize target and polar spans as continuous tokens. Polarity $\{\text{POS}, \text{NEU}, \text{NEG}\}$ relation tags are assigned to a word-pair if at least one of the words in a multiword span is present in the pair. If for a continuous target-opinon span pair, multiple polarity tags are predicted, the final tag is decided by majority.

The model includes an iterative prediction and inference strategy to capture interactions between mutually indicative information between spans. For instance, if a predicted word contains a target tag, it is less likely that it features in a polar span word-pair, and vice versa. We tested if iterative prediction and inference helped by optimizing the number of turns (0–3) as a hyperparameter in the optimization search. This showed us whether, for our task, polar spans and target/aspect terms are mutually indicative as they are in the explicit ABSA datasets.

**Table 5.** The meaning of tags in the implicit sentiment-polarity triplet task.

| Tag | Meaning |
|-----|---------|
| T | two words in word-pair $(w_i, w_j)$ belong to the same target span. |
| P | two words in word-pair $(w_i, w_j)$ belong to the same polar span. |
| Pos | two words in word-pair $(w_i, w_j)$ belong to a target and polar span and they stand in positive implicit sentiment relation. |
| Neu | two words in word-pair $(w_i, w_j)$ belong to a target and polar span and they stand in neutral implicit sentiment relation. |
| Neg | two words in word-pair $(w_i, w_j)$ belong to a target and polar span and they stand in negative implicit sentiment relation. |
| $\varnothing$ | no relation for word-pair $(w_i, w_j)$ |

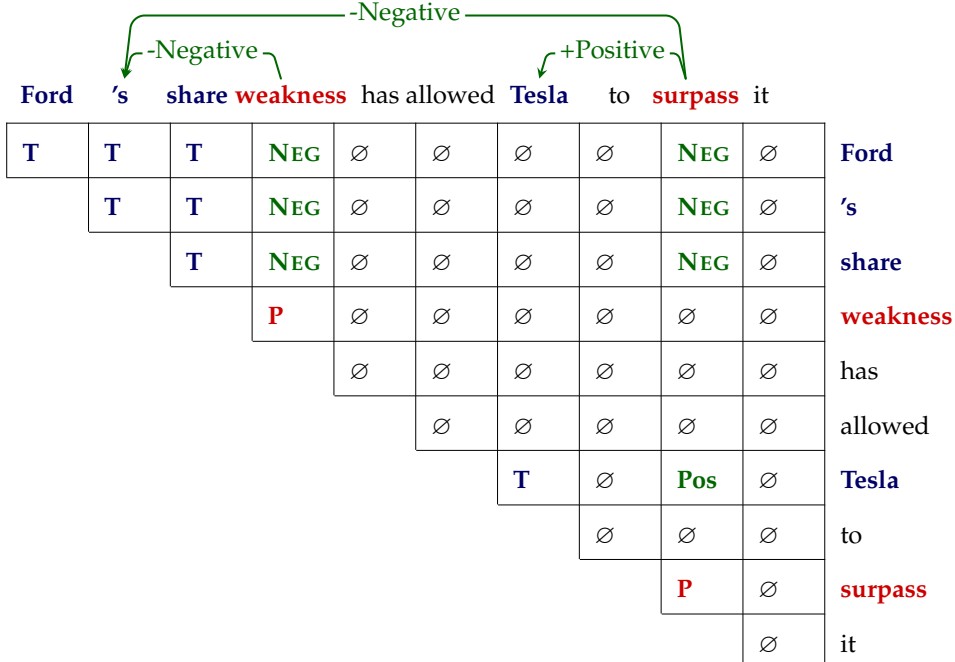

**Figure 6.** A grid-tagging example for the triplet extraction task. (Reproduced from Wu et al. [15] with a SENTiVENT corpus sample).

The choice of the GTS model was motivated by its state-of-the-art performance in an end-to-end model and its ability to encode one-to-many target-opinion relations, which fits our annotations. The concepts of a sentiment target and aspect are not identical, as aspects are usually predefined into specific categories, which leads to less lexical diversity in the terms. However, the grid-tagging approach remains a good fit for our implicit TBSA task.

GTS allows for any word vector encoding of the input sentence containing the triplet word pairs. The results showed that BERT encoding consistently had better performance than CNN and LSTM encoders, so we limited the tested encoding architectures to pretrained transformers. Because the GTS algorithm is more computationally intensive in word-pair tagging than the previous sequence-based experiments, we only tested BERT, RoBERTa, and FinBERT$_{TRC2+FP}$ encoders, as these obtained the best results in both previous experiments. We applied the same hyperparameter search model selection and holdout test procedure as in previous experiments; however, in line with [15], we retrained the best hyperparameterized model on the holdin (i.e., dev train + dev evaluation set) and tested on the holdout set five times, and averaged the final holdout score.

## 3. Results

### 3.1. Coarse-Grained Implicit Polarity Classification

For gold polar expression implicit polarity classification, we selected the best model on the basis of accuracy. Table 6 reports macro-averaged precision, recall, F1-score, and accuracy (eq. micro-avg F1). The best encoder model on the holdout test-set was RoBERTa$_{Large}$ without the addition lexicon features; RoBERTa with economic and general domain lexicons also performed best on the devset but showed overfitting. Out of the base-size models, we observed that RoBERTa$_{Base}$ is closely followed by DeBERTa$_{Base}$ in accuracy. Adding lexicon features often improves accuracy and F1 scores; however, there is a tradeoff in recall: lexicons overall improve precision but decrease recall.

The three top-scoring models did not make statistically significant differing predictions from the winning model according to McNemar's test. Of the models that produced statistically significant differing predictions, FinBERT$_{TRC2-FinPhrase}$ scored the highest. Adding a combination of domain-specific and general lexicon features improved accuracy performance by 1.143% on average across models, while domain-specific lexicons only improved accuracy by 0.714%. While the improvement is not substantial, adding lexicons does improve performance and can thus be recommended for this task.

**Table 6.** Implicit sentiment polarity classification results of gold polar expressions on development set (dev) and holdout set (test). Precision (P), recall (R), andd F$_1$-score (F$_1$) percentages are macro-averaged. Winning models selected on dev accuracy after optimizing hyperparameters for each architecture ablation. Accuracy (A) is reported with the *p*-value of McNemar's significance test of predictions with respect to the winning model. Boldface type indicates highest score overall and underlining indicates the highest score within encoder ablation.

| Model w/Lexicons | P | | R | | F$_1$ | | A | | *p* |
|---|---|---|---|---|---|---|---|---|---|
| | Dev | Test | Dev | Test | Dev | Test | Dev | Test | |
| FinBERT$_{TRC2+FP}$ [110] | 66.1 | 60.8 | 60.7 | 56.2 | 61.9 | 56.3 | 77.3 | 71.2 | <0.001 |
| + econ. | 68.8 | **72.2** | 58.0 | 54.0 | 59.1 | 52.5 | 77.8 | 73.2 | 0.007 |
| + econ.+general | 67.3 | 63.0 | 62.2 | 57.9 | 63.7 | 58.4 | 78.3 | 73.3 | 0.022 |
| BERT$_{Base}$ [106] | 66.4 | 57.3 | 63.8 | 55.4 | 64.1 | 54.6 | 78.2 | 68.3 | <0.001 |
| + econ. | 67.8 | 62.8 | 64.6 | 59.0 | 65.1 | 58.5 | 78.4 | 71.3 | <0.001 |
| + econ.+general | 66.0 | 59.8 | 64.0 | 57.9 | 64.5 | 57.3 | 78.3 | 71.2 | <0.001 |
| BERT$_{Large}$ [106] | 73.1 | 58.8 | 62.9 | 56.2 | 64.6 | 56.0 | 80.5 | 73.2 | 0.01 |
| + econ. | 69.1 | 61.8 | 69.7 | 61.6 | 69.3 | 61.5 | 79.1 | 72.8 | 0.004 |
| + econ.+general | 67.4 | 62.0 | 62.9 | 58.0 | 64.4 | 58.1 | 79.1 | 74.0 | 0.055 |
| FinBERT$_{FinVocab}$ [111] | 65.9 | 56.0 | 58.4 | 52.8 | 59.8 | 52.0 | 77.4 | 69.8 | <0.001 |
| + econ. | 65.0 | 58.2 | 60.2 | 56.1 | 61.9 | 56.2 | 76.0 | 71.6 | <0.001 |
| + econ.+general | 70.4 | 55.4 | 58.7 | 53.2 | 58.9 | 50.6 | 77.1 | 69.9 | <0.001 |
| DeBERTa$_{Base}$ [109] | 67.2 | 59.5 | 64.3 | 58.4 | 65.2 | 57.9 | 78.7 | 71.6 | <0.001 |
| + econ. | 66.8 | 62.7 | 63.0 | 57.7 | 63.5 | 57.7 | 78.9 | 72.5 | 0.001 |
| + econ.+general | 68.4 | 70.4 | 60.9 | 58.2 | 61.9 | 58.9 | 78.7 | 74.9 | 0.335 |
| RoBERTa$_{Base}$ [107] | 66.1 | 58.4 | 59.2 | 55.9 | 59.2 | 54.0 | 78.8 | 74.5 | 0.124 |
| + econ. | 67.7 | 63.8 | 66.1 | 61.5 | 66.7 | 61.5 | 79.4 | 74.4 | 0.135 |
| + econ.+general | 69.0 | 63.5 | 66.8 | 62.6 | 67.7 | 62.8 | 79.3 | 75.0 | 0.409 |
| RoBERTa$_{Large}$ [107] | 68.4 | 65.8 | 65.3 | 63.0 | 66.2 | 63.2 | 80.2 | 77.5 | 0.105 |
| + econ. | 68.6 | 63.8 | 66.4 | 62.6 | 67.4 | 63.0 | 79.7 | 75.4 | 0.579 |
| + econ.+general | **73.2** | 61.9 | 66.5 | 58.8 | 68.5 | 58.3 | **81.4** | 75.9 | - |

Table 7 shows the performance of the winning model across implicit polarity labels; we can see that good results are obtained for positive (83.7% F$_1$) and negative polarity, and that neutral lags far behind (15.5% F$_1$). This is expected due to the nature of the neutral tag, which is annotated when implicit polarity is ambiguous or no convincingly positive or negative sentiment can be confidently asserted. It is also the class with the least attestations, introducing issues of data imbalance and scarcity. Figure 7 shows that the neutral label is by far most often confused with positive instances. Overall, we see that with an accuracy of 77.5% and a macro-F$_1$ of 63.2%, implicit polarity classification is a challenging task, even when given known sequences of polar expressions.

**Table 7.** Precision (P), recall (R), and F$_1$-score percentages, and support by label for (a) gold polar expression and (b) clause-based implicit polarity classification.

| | **(a) Gold Polar Expressions** | | | | **(b) Clause-Based** | | | |
|---|---|---|---|---|---|---|---|---|
| | **P** | **R** | **F$_1$** | **Support** | **P** | **R** | **F$_1$** | **Support** |
| Positive | 79.6 | 88.2 | 83.7 | 727 | 82.1 | 62.7 | 71.1 | 577 |
| Neutral | 32.6 | 10.1 | 15.5 | 148 | 31.2 | 20.2 | 24.5 | 124 |
| Negative | 73.4 | 78.2 | 75.7 | 363 | 68.4 | 60.9 | 64.4 | 266 |
| None | - | - | - | - | 54.6 | 86.8 | 67.0 | 355 |

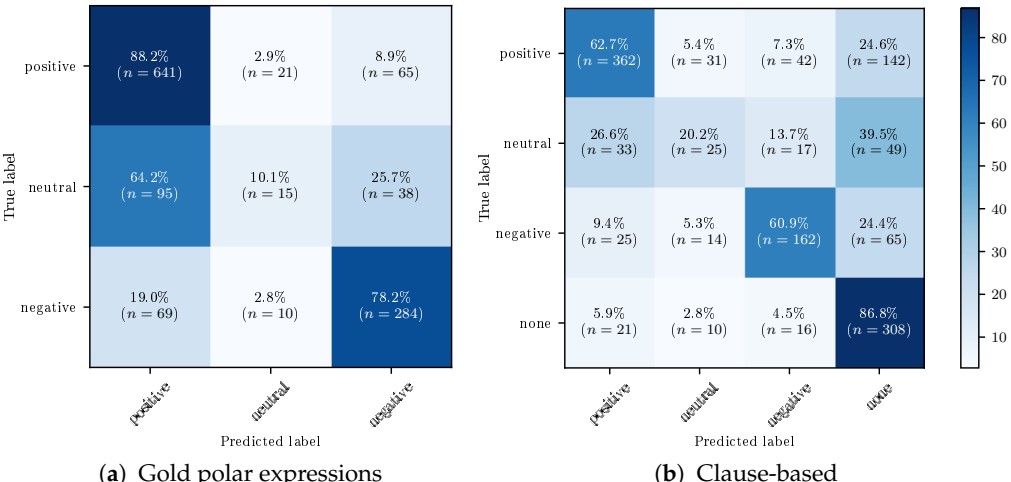

**(a)** Gold polar expressions      **(b)** Clause-based

**Figure 7.** Confusion matrices for (**a**) Gold polar expression and (**b**) Clause-based implicit polarity classification.

For the clause-based experiments, we selected the winning model based on macro-F$_1$ as macro-averaging performance over the four labels {None, Pos, Neu, Neg}. Table 8 shows RoBERTa$_{Large}$ again obtained the best scores on the test sets, however now adding economic sentiment lexicon features results in the best model with an F$_1$ of 56.7% and an accuracy of 64.8%. Of all the base-size models, RoBERTa and DeBERTA perform best.

Adding economic lexicon features improved F$_1$ scores by 1.2% across all models, while adding combined general and economic lexicons decreased performance by 1.39%. All tested models performed worse on the holdout test set when general domain lexicons were added. From this, we conclude that adding in-domain lexicon features increases the model's ability to identify implicit polar expressions, while adding general domain lexicons increases the likelihood of false positives in detection. Lexically explicit sentiment might be present, but it is not relevant to detecting economic implicit sentiment.

Table 7 shows that the neutral label has the lowest performance at 24.5%, while positive, negative, and none categories obtain acceptable scores of 71.1%, 64.4%, and 67.0%, respectively. Figure 7 shows that the majority of misclassified neutral instances are confused with the none label, followed by the positive class. Again, this is unsurprising due to the inherent difficulty in distinguishing the neutral category with the absence of a label. Across labels, detection of polar expressions remains difficult, with the polarity labels being most often confused with the none category, indicating misses in detection.

**Table 8.** Clause-based implicit sentiment results on the development set and holdout test set for winning models after optimizing the hyperparameters for each architecture. Precision (P), recall (R), and $F_1$-score ($F_1$) percentages are macro-averaged. Accuracy (A) is reported with the *p*-value of McNemar's significance test of predictions with respect to the best model (RoBERTa$_{Large}$ + econ.). Boldface type indicates highest score overall and underlining indicates the highest score within encoder ablation.

| Model w/ Lexicons | P | | R | | $F_1$ | | A | | $p$ |
| --- | --- | --- | --- | --- | --- | --- | --- | --- | --- |
| | Dev | Test | Dev | Test | Dev | Test | Dev | Test | |
| FinBERT$_{TRC2+FP}$ [110] | 56.1 | 53.7 | 56.3 | 53.9 | 55.9 | 52.2 | 63.1 | 60.0 | <0.001 |
| + econ. | 56.5 | 53.6 | 57.0 | 54.8 | 56.6 | 53.4 | 64.9 | 60.2 | <0.001 |
| + econ.+general | 55.0 | 52.5 | 55.7 | 53.1 | 55.3 | 52.0 | 62.8 | 60.4 | <0.001 |
| BERT$_{Base}$ [106] | 57.0 | 54.2 | 57.9 | 53.9 | 57.4 | 53.1 | 64.8 | 62.5 | 0.059 |
| + econ. | 57.2 | 55.3 | 56.6 | 54.6 | 56.8 | 53.8 | 65.2 | 62.3 | 0.04 |
| + econ.+general | 56.8 | 53.2 | 57.1 | 53.7 | 56.9 | 52.9 | 64.9 | 61.6 | 0.012 |
| BERT$_{Large}$ [106] | 57.5 | 54.8 | 57.7 | 55.8 | 57.6 | 54.6 | 65.8 | 62.0 | 0.018 |
| + econ. | 57.4 | 56.4 | 58.1 | 57.1 | 57.5 | 55.6 | 65.6 | 62.8 | 0.086 |
| + econ.+general | 59.5 | 55.1 | 57.2 | 52.3 | 57.7 | 50.3 | 68.0 | 61.3 | 0.002 |
| FinBERT$_{FinVocab}$ [111] | 60.0 | 53.0 | 52.8 | 52.6 | 53.4 | 49.7 | 64.6 | 61.7 | 0.009 |
| + econ. | 59.3 | 54.2 | 55.2 | 54.1 | 55.4 | 51.8 | 64.6 | 62.5 | 0.056 |
| + econ.+general | 57.2 | 51.7 | 54.5 | 52.2 | 54.5 | 50.1 | 65.6 | 62.6 | 0.084 |
| DeBERTa$_{Base}$ [109] | 61.0 | 53.8 | 57.8 | 53.7 | 58.5 | 52.5 | 67.6 | 63.3 | 0.214 |
| + econ. | 59.6 | 56.7 | 57.5 | 56.0 | 58.1 | 55.2 | 66.6 | 63.7 | 0.34 |
| + econ.+general | **64.7** | 52.9 | 56.5 | 52.4 | 57.7 | 50.1 | 68.5 | 64.0 | 0.493 |
| RoBERTa$_{Base}$ [107] | 60.7 | 57.8 | 58.0 | 55.3 | 58.7 | 55.1 | 67.5 | **66.2** | 0.26 |
| + econ. | 58.0 | 56.2 | 58.3 | 56.6 | 58.1 | 55.7 | 66.3 | 63.7 | 0.373 |
| + econ.+general | 61.1 | 57.6 | 58.3 | 56.0 | 58.4 | 54.2 | 69.0 | 65.5 | 0.541 |
| RoBERTa$_{Large}$ [107] | 59.5 | 56.7 | 60.5 | 57.8 | 59.5 | 56.7 | 68.1 | 64.8 | 0.944 |
| + econ. | 61.9 | 59.1 | 59.7 | 57.6 | 60.3 | 56.8 | 68.2 | 64.8 | - |
| + econ.+general | 61.3 | 56.0 | 59.0 | 55.6 | 59.6 | 54.6 | 68.4 | 63.5 | 0.232 |

Overall, the FinBERT$_{TRC2+FP}$ model, which was further pretrained on and subsequently fine-tuned on the FinancialPhrasebank sentiment classification task, showed acceptable performance. Here, we can see the advantages of sequential fine-tuning on similar tasks, as the FinancialPhrasebank task is a sentence-level classification task for positive, neutral, and negative economic sentiment, followed by our own fine-tuning. However, the in-domain FinBERT$_{FinVocab}$ model, which was pretrained on financial text, performed on the lower-end, in-line with general-domain BERT. The presupposed advantages of similar-domain pretraining and vocabulary could be moot due to financial news being more similar to general language than the corporate reports, earning call transcripts, and analyst reports of the FinBERT$_{FinVocab}$ dataset.

### 3.2. Fine-Grained Triplet Results

For the fine-grained triplet task, we tested the grid-tagging scheme (GTS) model [15] with several transformer-encoders. Table 9 shows the performance of the final triplet task, which required extracting the exact target and polar spans, and classifying polarity. We tested GTS on our dataset and the joined explicit sentiment sub-datasets of Wu et al. [15] for comparison of performance with the state-of-the-art method in explicit sentiment processing with our implicit sentiment task.

The best-performing model of our dataset is RoBERTa with an $F_1$-score of 21.7%. The same approach on the explicit sentiment triplets results in a score of 75.5%, in line with the scores reported on the subdatasets in the original work by Wu et al. [15]. From this substantial drop in performance across task, we conclude that the existing state-of-the-art method in PLM models cannot handle the complexity of our implicit sentiment task. Table 10 shows the performance on target/aspect and polar span extraction. Target term extraction obtains an $F_1$-score of 55.3%, while the implicit polar span extraction obtains 38.9%, indicating that extracting the token-span of targets of implicit sentiment is easier than extracting the exact spans of polar expressions. This is expected as targets are better lexically delineated than implicit polar spans. The same effect was not observed in explicit

sentiment where the lexical diversity of both the domain and terms is much lower than in polar sentiment analysis.

**Table 9.** Fine-grained token-level implicit sentiment results on the development set and holdout test set for winning models after optimizing hyperparameters for each architecture. Precision (P), recall (R), and $F_1$-score ($F_1$) percentages are micro-averaged.

| SENTiVENT (Ours) | Triplet P | | Triplet R | | Triplet $F_1$ | |
|---|---|---|---|---|---|---|
| | **Dev** | **Test** | **Dev** | **Test** | **Dev** | **Test** |
| BERT$_{Base}$ [106] | 20.2 | 19 | 17.3 | 14.3 | 18.6 | 16.3 |
| BERT$_{Large}$ [106] | 19.9 | 19.3 | 19.9 | 17.6 | 19.9 | 18.2 |
| DeBERTa$_{Base}$ [109] | 25.8 | 21.1 | 18.3 | 17 | 21.4 | 18.7 |
| FinBERT$_{TRC2+FP}$ [110] | 20 | 20.4 | 19 | 17.4 | 19.4 | 18.8 |
| RoBERTa$_{Base}$ [107] | 24.1 | **24.4** | **21.4** | 19.7 | **22.7** | **21.7** |
| RoBERTa$_{Large}$ [107] | **28.3** | 23.1 | 17.8 | **19.7** | 21.9 | 21.1 |
| Explicit Wu et al. [15] | | | | | | |
| RoBERTa$_{Base}$ [107] | **80.0** | **76.1** | 74.4 | **74.9** | **77.1** | **75.5** |
| RoBERTa$_{Large}$ [107] | 73.9 | 75.3 | **76.6** | 68.6 | 75.2 | 71.7 |

**Table 10.** Fine-grained aspect and polar span extraction results on holdout test set for winning models after optimizing hyperparameters for each architecture. Precision (P), recall (R), and $F_1$-score ($F_1$) percentages are micro-averaged.

| SENTiVENT (Our) | Target Span | | | Polar Span | | |
|---|---|---|---|---|---|---|
| | **P** | **R** | **$F_1$** | **P** | **R** | **$F_1$** |
| BERT$_{Base}$ [106] | 49.6 | 47.1 | 48.2 | 35 | 33.8 | 34.3 |
| BERT$_{Large}$ [106] | 49.3 | 51.4 | 50.2 | 32 | 38.5 | 34.7 |
| DeBERTa$_{Base}$ [109] | 50.5 | 53.1 | 51.6 | 35.3 | 36.3 | 35.6 |
| FinBERT$_{TRC2+FP}$ [110] | 51.3 | 51.8 | 51.5 | 34.6 | 37.4 | 35.9 |
| RoBERTa$_{Base}$ [107] | **55.0** | 55.6 | **55.3** | **39.0** | 38.9 | **38.9** |
| RoBERTa$_{Large}$ [107] | 50.3 | **59.5** | 54.4 | 35.9 | **40.1** | 37.8 |
| Explicit Wu et al. [15] | | | | | | |
| RoBERTa$_{Base}$ [107] | **85.3** | **89.2** | **87.2** | **87.1** | **88.9** | **88.0** |
| RoBERTa$_{Large}$ [107] | 85.1 | 86.1 | 85.6 | 85.5 | 88.7 | 87.1 |

## 4. Discussion

Concerning the coarse-grained experiments, the scores for implicit polarity classification of gold polar expressions (76% accuracy) indicate that assigning implicit polarity is a challenging task but otherwise viable. In the clause-based experiments, acceptable performance (57% macro-F1 and 65% accuracy) was obtained, showing the feasibility of detection of clause-level implicit sentiment.

There are few similar works with which we can compare coarse-grained performance for implicit polarity classification. Compared to similar coarse-grained polarity tasks [25,125,126], our coarse-grained performance is lower with less complex but similar PLM and attention-based models. This indicates increased difficulty in our dataset compared with other coarse-grained sentiment tasks. This is likely due to the more lexically open domain of our corpus: while the domain is limited to company-specific news, it encompasses a large range of topics and thus has larger lexical diversity than the studies mentioned previously. In ensuring representativeness of economic news, many different sectors and both consumer-oriented and more technical reporting were included. Additionally, unlike many other datasets, the polarity labels are also not balanced, with the neutral label being a minority class.

Regarding the fine-grained triplet experiments, the large performance gap of our dataset compared to the explicit sentiment review dataset of Wu et al. [15] showed that the current state-of-the-art model in explicit sentiment is not sufficient for our SENTiVENT dataset. Explicit ABSA is domain-constrained to one product category (e.g., laptops) with predefined aspect categories. The large difference in performance between the explicit task

and implicit economic sentiment task shows that state-of-the-art lexically based methods that have proven successful for fine-grained sentiment analysis on review data will not suffice for implicit sentiment tasks in economic newswire text. The increased difficulty of the SENTiVENT implicit sentiment dataset vs. explicit opinionated, user-generated content lies in the lexical openness of our annotations with less syntactically constrained span annotations (target and polar spans are not limited to certain types of phrases or syntactical boundaries). We see this reflected in the span extraction results, where implicit polar spans are much more difficult to identify than explicit spans or even their target spans (Table 10). Implicit sentiment is much semantically abstracted and hence less well lexically defined than explicit sentiment.

Nearly all winning hyperparameters for the GTS model did not include the iterative prediction and inference strategy (i.e., $t = 0$), whereby it captures mutually indicative information between target and polar spans in the explicit task. The lack of improvement indicates that, compared to explicit sentiment, the more lexically open targets and polar spans in our dataset contain less mutually indicative clues. This limits the applicability of unified modeling approaches to this dataset.

Regarding recommendations for further research and downstream applications (such as stock movement prediction or sentiment index aggregation) for our dataset, the inherently ambivalent neutral instances can often be omitted. This would greatly improve performance, as neutral predictions represented the largest number of confused labels and had the lowest performance of all classes. Regarding encoders, RoBERTa is the most promising, being robust across tasks, outperforming other in-domain pretrained encoders. We also observed that further in-domain pretraining and sequential multitask fine-tuning produced performance improvements over the vanilla Bert model. Although we noticed that adding lexicons did not substantially increase scores for implicit sentiment, we recommend using in-domain without general domain lexicons as it produced a small but consistent improvement across coarse-grained task. Sentiment lexicons might also be beneficial in applications where system precision is more important than recall.

In order to gain further insight into the errors produced by the different systems, we performed a qualitative error analysis on the output of the best model for each coarse-grained task. As current fine-grained performance is too low, an error typology would provide little useful information regarding avenues of improvement. We identified and manually assigned error types to a subset (50% of gold and 41% of clause-based experiments) of classification errors on the test set. In Table 11, we list different error categories and their frequency for both coarse-grained tasks.

**Table 11.** Manual error category frequencies for gold polar expression polarity and clause-based polarity test set errors.

|  | Gold Polar Expressions | Clause-Based |
| --- | --- | --- |
| Unusual Language | 8.0% (n = 10) | 22.9% (n = 41) |
| Clues in global context | 37.6% (n = 47) | 28.5% (n = 51) |
| Ignored Strong lexical Cue | 12.0% (n = 15) | 18.4% (n = 33) |
| preprocessing | 32.8% (n = 41) | 29.1% (n = 52) |
| Plausible prediction | 55.2% (n = 69) | 39.7% (n = 71) |

"Unusual language" indicates highly idiomatic, creative language (e.g., sayings and figures of speech, e.g., Examples 1 and 2), or a high density of infrequent lexical items (usually sector-specific jargon, e.g., Example 3). This causes errors due to data scarcity.

1. *"you hear a fair amount of Buck Rogers-sounding futurism"* | True: Positive | Pred: None
2. *"Where's The Tylenol?"* | True: negative | Pred: positive
3. *"comes in a high-performance SRT package or a Trackhawk edition with a monstrous Hemi V-8 engine"* | True: Positive | Pred: None

"Clues in global context" is another frequent error category, meaning that sentiment must be derived from contexts outside of the input string, in other clauses, or sentences throughout the document, e.g.,

1. *"biosimilars impacting the company's future growth"* ∣ True: Negative ∣ Pred: Positive → *"impacting"* can be both positive and negative, that lexical ambiguity is resolved in the previous sentences where it is made clear that *"biosimilars"* are competitors.
2. *"Ford CFO Bob Shanks noted that 2018 earnings were likely to be in the range of $1.45–$1.70"* ∣ True: Negative ∣ False: Positive. Later in the article, higher expected earnings for that year are discussed and missing earnings predictions is negative; however, our coarse sample does not include this context.

This highlights the need for global context modeling as well deep contextual natural language understanding across the whole document. Example 2 also highlights the need for temporal and numerical reasoning in financial sentiment tasks: economic performance is often inferred from comparing metrics to past performance or to the general market trend. Market trends are not always mentioned in articles and often the author assumes common-ground knowledge. Modeling this would require document-external data sources and modeling of market performance and participants.

"Ignored strong lexical cue" includes instances where a strong lexical indicator of positive or negative sentiment was present. Usually there are several labeled instances containing this lexical cue in the dataset. The model failed to learn the strong sentiment association of this word.

1. *"Amazon's **rally**, the market is still **underestimating** the company"* ∣ True: Positive ∣ Pred: Negative
2. *"sees **growth accelerating** for the full-year"* ∣ True: Positive ∣ Pred: None

"Preprocessing" from fine-grained annotations to coarse-grained segments sometimes introduces errors. Often, lexical clues are omitted by addition or omission of positive or negative indicators when splitting clauses, or going from discontinuous to continuous spans (e.g., Example 2). Errors are also caused by the majority voting mechanism in clause-based preprocessing, viz. when multiple polar expressions are present in a clause the, most common polarity is chosen. The majority polarity is not always the most salient (Example 1).

This error category is a weakness of our fine-to-coarse-grained transformation approach.

1. *"Apple's **growth** has been **lagging the growth of the market**, with IHS reporting two percent **growth** in a similar period"* → Positive, Negative, Positive → True: Positive by majority vote ∣ Negative (plausibly correct as the negative sentiment is most salient in this segment).
2. "organic sales growth projections for this year, based on management' s comments, suggested ∣ Negative ∣ Positive ∣ from original "**organic sales growth projections** for this year , based on management's comments, suggested the **momentum** seen in the fourth quarter **wouldn't continue** at the same pace"

For both tasks, "plausible prediction" is the most frequent error category. A plausible prediction occurs when the predicted label can reasonably be assigned to the instance, viz. these instances could be considered correct. This often co-occurs with all of the above error types when the input text string in isolation can plausibly be assigned the polarity. This happens frequently with neutral vs. positive or negative confusion where there is inherent ambivalence. Our annotators have global context, which can shift commonly connotationally positive and negative events toward neutral events when the context introduces ambivalence, as discussed above. Often this also co-occurs due to a preprocessing error where immediate context clues are removed, which resulted in different polarity, but now are actually correct. For the clause-based experiments, these also include plausibly spurious predictions: clauses where no gold label is present but should be.

1. "*shift toward autonomous cars*" | True: Neutral | Pred: Positive
   → Plausible due to inherent ambivalence of *neutral*.
2. "*Amazon could acquire a larger entity*" | True: Neutral | Pred: Positive → Plausibly positive as annotator chose neutral due to modality.
3. "*Roughly a third of the additions last year were revisions to existing reserves*" | True: None | Pred: Neutral
   → A plausible spurious prediction where a potential polar event annotation was not annotated.
4. "*he helped fund a Steinhoff capital-raising*" | True: Negative | Pred: Pos
   → From a global context, it is explained that Steinhoff was a fraudulent bank; in isolation, being funded is generally positive.

## 5. Conclusions

In this paper, we presented the SENTiVENT dataset with fine-grained annotations for implicit sentiment in English economic news. Our work focused on two research strands in the domain of natural language processing: implicit sentiment detection and fine-grained sentiment analysis, which both have mainly been researched in the framework of user-generated content with strongly lexicalized opinion. As many of the existing approaches to financial sentiment analysis remain coarse-grained, our work fills the need for a manually labeled implicit and explicit economic dataset, and we think that this rich resource will fuel future research on fine-grained target-based sentiment analysis of full news articles. We validated the annotation scheme with an inter-annotator agreement study, proving the high quality of the annotations.

To assess the feasibility of implicit sentiment detection in economic news, we presented three sets of experiments ranging from gold polar expression polarity classification and clause-based implicit polarity classification to end-to-end extraction of <polar span, target span, polarity> triplets. We found acceptable performance on coarse-grained experiments, demonstrating the feasibility of detecting positive, neutral, and negative implicit sentiment polarity. The fine-grained triplet task, however, remains a challenge even to current unified representation-learning methods. We showed that a state-of-the-art model for fine-grained sentiment triplet extraction, which has proven successful on a benchmark explicit sentiment dataset, is not easily portable to our implicit investor sentiment task. Error analysis showed a large lexical variety within polar expressions typical of implicit sentiment in more objective text genres such as economic news. This implies the need for methodologies that exceed flat lexical inputs to alleviate data scarcity for extracting implicitly polar spans.

Hence, in future work, we plan to experiment with multitask learning approaches [127,128] to take advantage of existing similar resources such as FiQA [83], SentiFM [88], and FinancialPhrasebank [80]. Regarding modeling approaches, we plan to include syntax-level [129,130] and external sentiment resources by information fusion [131] because lexical feature learning is currently not sufficient for implicit fine-grained triplet extraction. We will also create a financial sentiment lexicon derived from this dataset as the fine-grained annotations are a better fit for lexicon-learning [18,118] than most existing coarse-grained approaches.

**Author Contributions:** Conceptualization, G.J. and V.H.; methodology, G.J. and V.H.; software, G.J.; validation, G.J.; formal analysis, G.J.; investigation, G.J.; resources, G.J.; data curation, G.J.; writing—original draft preparation, G.J.; writing—review and editing, G.J. and V.H.; visualization, G.J.; supervision, V.H.; project administration, G.J. and V.H.; funding acquisition, G.J. and V.H. All authors have read and agreed to the published version of the manuscript.

**Funding:** This research was carried out as part of a Ph.D. fellowship on the SENTiVENT project, funded by the Research Foundation—Flanders.

**Data Availability Statement:** All experiment replication data and source code is available through the scientific repository at osf.io/cbjta (accessed on 7 September 2021) [132]. All data in coarse- and fine-grained formats is made available for replication. This includes all source code of the model

architectures, hyperparameter search, parsing and preprocessing, and other utility scripts for result table and figure generation. The original WebAnno files, including all annotated events, sentiment, and factuality features will be added to this repository at the SENTiVENT project's end.

**Acknowledgments:** Thanks to Wu et al. [15] for releasing the source code to the GTS triplet extraction model.

**Conflicts of Interest:** The authors declare no conflict of interest. The funders had no role in the design of the study; in the collection, analyses, or interpretation of data; in the writing of the manuscript, or in the decision to publish the results.

## Abbreviations

The following abbreviations are used in this manuscript:

| | |
|---|---|
| SA | Sentiment Analysis |
| ABSA | Aspect-Based Sentiment Analysis |
| CRM | Client-Relation Management |
| TBSA | Target-Based Sentiment Analysis |
| PLM | Large-scale pretrained Language Model |
| GTS | Grid-Tagging Scheme model by Wu et al. [15] |
| FP | FinancialPhrasebank dataset by Malo et al. [80] |
| TRC2 | Reuters Financial news corpus Araci [110] |
| ML | Machine Learning |
| IAA | Inter-Annotator Agreement |
| NLP | Natural language processing |
| IE | Information Extraction |
| AE | Aspect span Extraction |
| PE | Polar span Extraction |
| SC | Sentiment Classification |
| AESC | Aspect term extraction and sentiment classification |
| PESC | Polar span Extraction & Sentiment Classification |
| TESC | Target span Extraction & Sentiment Classification |
| P | Precision score |
| R | Recall score |
| F1 | $F_1$-score |
| A | Accuracy |

## Appendix A. Agreement for Overlapping Token-Spans

We compute agreement metrics by using matching for sentiment expression based on span overlap. Since events and sentiment are semantic categories with no set syntactic or lexical boundary rules, there is some variation in boundaries of token spans. Other work on ABSA resources perform agreement scoring at the sentence-level [33,99] or clause-level [100], thus circumventing the issue of matching token-level annotations, but they also lose granularity. Our span-matching approach, presents a more rigorous attempt at unitizing spans for computing agreement.

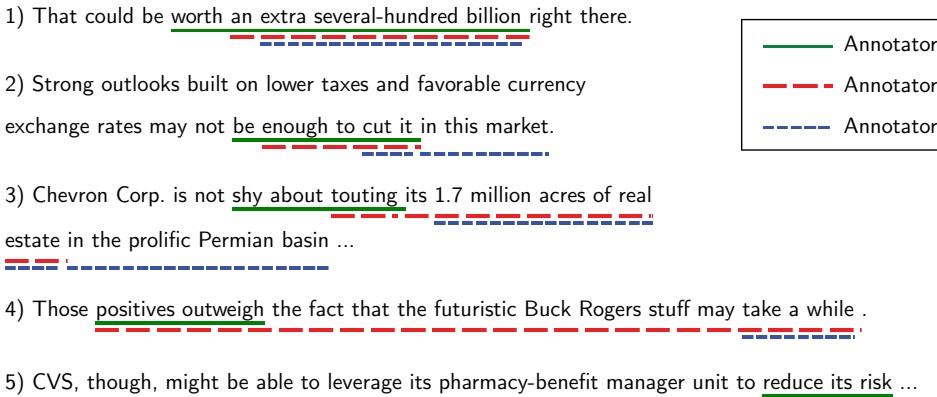

**Figure A1.** Examples of token span overlap for polar spans.

Figure A1 show typical examples of annotations where boundaries do not match exactly, but the core meaning remains valid. Applying exact span agreement in which only exactly matching token spans would underestimate the agreement of labels.

We first unitize the text by matching overlapping spans of annotations into groups of annotations and subsequently compute agreements metrics of those groups. In other work where overlapping span alignment is used [98,133], this is done pair-wise between annotators, with the final agreement score being the mean of each annotator pair. However averaging of pair-wise scores does not allow calculating the mean expected agreement distribution of labels across all annotators in Krippendorf's alpha. Hence we make alignment groups for all three annotators at once in order to use generalized agreement metrics which allow for more than two annotators. Token overlap is quantified based on the dice-coefficient (equivalent to $F_1$-score) of overlapping spans. When one annotator has multiple candidates that overlap with others, the selection is made based on maximum dice-coefficient capturing which has the best overlap. The other non-matched annotations are then placed in their own alignment group. In example 5 in Figure A1, the longer "*leverage its pharmacy-benefit manager unit to reduce its risk*" by Annotator 2 is aligned with "*leverage its pharmacy-benefit manager unit*" of Annotator 3 instead of "*reduce its risk*" because it has larger token overlap.

Annotators may tag polar span annotations in the same exact place or with overlap if the polarity/event type is different. As an edge-case this self-overlap can be exact, as in Figure A2, in which case our span matching approach has two equivalent candidates with the same dice-score. When this happens, we disambiguate by matching on the label. If no label corresponds to the other annotators, one is randomly selected for alignment. The span that does not match is then placed into its own unmatched annotation group.

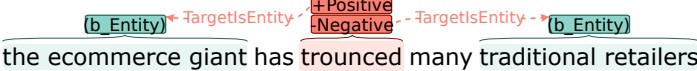

**Figure A2.** Example of exact span overlap of the same annotator.

## Appendix B. Pretrained Model References

We provide the location of weights used in our dataset. Most pretrained models where obtained from the Huggingface/transformers [134] models repository. The replication repository contains exact model references and dates of access in source code.

| Name & Reference | Location of Used Weights |
|---|---|
| BERT$_{Base}$ [106] | https://huggingface.co/bert-base-cased |
| BERT$_{Large}$ [106] | https://huggingface.co/microsoft/bert-large-cased |
| DeBERTa$_{Base}$ [109] | https://huggingface.co/microsoft/deberta-base |
| FinBERT$_{TRC2+FP}$ [110] | https://huggingface.co/ProsusAI/finbert |
| FinBERT$_{FinVocab}$ | https://github.com/yya518/FinBERT |
| RoBERTa$_{Base}$ [107] | https://huggingface.co/roberta-base |
| RoBERTa$_{Large}$ [107] | https://huggingface.co/roberta-large |

## Appendix C. Classification Head

We use no pooling of hidden representations of the final transformer layers as input for the classification head, i.e., we only use the final hidden representation of the transformer model. For the gold polarity and clause-based experiments, we change the text sequence classification heads of the models to consistently use RoBERTa's head as implemented by default in HuggingFace/transformers [134]. In preliminary testing this classification head performed slightly better with both BERT and RoBERTa. This consists of Dropout $\rightarrow$ Dense linear layer $\rightarrow$ Tanh activation $\rightarrow$ Dropout $\rightarrow$ Output layer. This was to make fine-tuning of the encoder architectures consistent and final classification not dependent on implementation differences in classification heads allowing direct comparison of pretraining model quality.

## Appendix D. Hyperparameter Optimisation

Table A1 shows the samplespaces used in hyperparameter search. Of main interest are the lexicon feature groups, which can consist of the matching scores for positive, neutral, and negative sublexicons (*pos-neg-neu*), polarity scores derived by subtracting positive and negative scores (*polarity*), or all sublists combined (including psychometric sublists of LIWC) with derived polarity scores (*combined*). We run hyperparameter optimization maximizing macro-F$_1$ score using a Bayesian search with hyperband stopping [103] for 128 runs. For this we use the parameter sweep functionality in WandB.ai's Python package Biewald [135].

**Table A1.** Hyperparameter searchspace.

| Hyperparameter | Base Model | Large Model |
|---|---|---|
| Learning rate | $lr \in [4e^{-5}, 8e^{-5}]$ | $lr \in [4e^-5, 1e-4]$ |
| Coarse-grained | | |
|     Batch size | $b \in [32, 64]$ | $b \in [16, 32]$ |
|     Lexicon sublist features | $F \in \{\text{pos-neg-neu, polarity, combined}\}$ | |
| Fine-grained GTS | | |
|     Iterative inference steps | $t \in \{0, 1, 2, 3\}$ | |

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
