# Peer review of "Fine-Grained Implicit Sentiment in Financial News: Uncovering Hidden Bulls and Bears"

_electronics, doi:10.3390/electronics10202554_

Round 1

Reviewer 1 Report

- There is a need to revise the paper title, and the current title does not portray the actual meaning of the paper. 

- The abstract is not well written. There is a need to revise with explicit contents of the abstract, i.e., the main issue, methods, results, and implication. The author(s) should provide a precise and focused abstract. 

- As a suggestion for improvement, the author(s) should not use the same Keywords as like Paper Title. It is encouraged to used different keywords which are not in the Paper title. It will enhance paper searchability after publication.

  • The introduction section is not well written. There are ambiguous statements and no clarity in the introduction section.
  • There is no roadmap at the end of the introduction that conveys the structure of the rest of the paper.
  • Quality of Communication: The paper needs further proofreading. I have tried to read the paper constructively, but I felt it suffers from poor writing. I, therefore, request the author(s) to pass the manuscript for professional proofreading. I suggest that a more careful investigation of prior literature can make this paper distinguishable. Linking this article with prior studies does not seem sufficient, which weakens the justification of incremental contributions.
  • The in-text citations and end list of references do not sufficiently correspond. Please cross-check and correct citations and references throughout the paper. 
  • The author(s) did cite the latest literature relevant to the target issue in this paper. The reviewer found that author(s) has cited only a few recently published papers in this article (Most of the cited articles are five years old). As a suggestion, the author(s) must cite new articles (latest literature) to make holistic discussion and sturdy paper with high readability
  • Author(s) should provide concluding statements rather than repetitive statements in the conclusion portion.
  • I hope that the comments provided can help in this regard.

Author Response

Reviewer 1:

We thank the reviewer for their comments and have made revisions for all but the title.
Their comments have surely improved the quality of the manuscript.
We have proofread the whole manuscript and fixed grammar and spelling errors, citations, argumentation and flow throughout.
We have rewritten the introduction and conclusion as requested.
Please find a more detailed response on a per remark basis below:

- There is a need to revise the paper title, and the current title does not portray the actual meaning of the paper. 
    - We respectfully disagree: The term ``Fine-grained'' commonly describes token-level annotations/tasks, for which there is a need in resources and approaches for implicit sentiment which this paper fills. Both target spans and sentiment spans are annotated and for the fine-grained experiments, we extracted these token-level span labels and target relations. So "fine-grained" is an apt and common characterization of the resource, task, and experiments.
    "Implicit sentiment": We provide event-implied investor sentiment as sentiment polarity which can be inferred by the reader through world-knowledge. This is a common definition of implicit sentiment and this is explained in the introduction and related research. We have elaborated this definition of implicit sentiment with a discussion of the examples and a revision of the introduction section. Possibly there is terminological confusion by another use of "implicit" in the term "implicit polarity classification": this is sometimes found in the ABSA literature when post/document-level polarity is classified, as in for instance Chen and Chen (2016). The term "implicit" is then a matter of annotation granularity and can include both fact-implied implicit and explicit sentiment. But that definition is narrow, is not concerned with how sentiment itself is defined, and is not the definition in the line-of-research we clearly set out in related research. This is all explained and framed in the paper w.r.t. previous research and definitions.

- The abstract is not well written. There is a need to revise with explicit contents of the abstract, i.e., the main issue, methods, results, and implication. The author(s) should provide a precise and focused abstract. 
    - We agree and have rewritten the abstract completely to specify the issue, methods, results, and findings.

- As a suggestion for improvement, the author(s) should not use the same Keywords as like Paper Title. It is encouraged to used different keywords which are not in the Paper title. It will enhance paper searchability after publication.
    - We changed the keywords to still be descriptive of the content and have no overlap with the title.

- The introduction section is not well written. There are ambiguous statements and no clarity in the introduction section.
    - We reworked the introduction section to be more clear and less ambiguous. We more clearly stated background, examples, and merits to make the introduction more focused.

- There is no roadmap at the end of the introduction that conveys the structure of the rest of the paper.
    - The submission in fact did contain a road-map with paper structure laying out which topics are discussed in which sections at the end of the introduction section.

- Quality of Communication: The paper needs further proofreading. I have tried to read the paper constructively, but I felt it suffers from poor writing. I, therefore, request the author(s) to pass the manuscript for professional proofreading.
    - We have further proofread the manuscript, improved writing and argumentation, and fixed grammar, spelling and typos throughout.

- I suggest that a more careful investigation of prior literature can make this paper distinguishable. Linking this article with prior studies does not seem sufficient, which weakens the justification of incremental contributions.
    - We have performed extensive search of all relevant, substantial resource creation research regarding fact/event-implied and fine-grained SA.
    The related research served its purpose in demonstrating similar fine-grained sentiment resources, historical definitions of implicit sentiment and to demonstrate financial applications.
    This section additionally demonstrates a clear need and gap in research for fact-implied implicit sentiment with fine-grained token-level annotations which we fill.
    If the reviewer is not specified with revisions would they mind specifying which specific topic or line-of-research needs expansion?

- The in-text citations and end list of references do not sufficiently correspond. Please cross-check and correct citations and references throughout the paper. 
    - We cross-checked citations and references and have fixed references where there was little correspondence.

- The author(s) did cite the latest literature relevant to the target issue in this paper. The reviewer found that author(s) has cited only a few recently published papers in this article (Most of the cited articles are five years old). As a suggestion, the author(s) must cite new articles (latest literature) to make holistic discussion and sturdy paper with high readability.
    - The work on implicit sentiment as fact-implied or reader-inferred polarity is sparse and indeed fairly old, we did manage to find some recently published work from the past months which has been added to the introduction.
    A vast amount of current work is indeed done in improving graph modeling on benchmark datasets such as the SemEval ABSA sets, but very little recent work has been done on event/fact-implied implicit sentiment that we are aware off. All relevant substantial works w.r.t. to resources and validation of manually labeled fine-grained, implicit sentiment have been added in the manuscript and have been discussed. Regarding target-based methods we included the most important datasets and much of the most recent modeling methods. We maintain a focus on corpus creation and validation for implicit sentiment and we feel no need to include every latest incremental modeling paper on common explicit ABSA benchmark subtasks. If Reviewer 1 feels we missed relevant research, we welcome and will add specific suggestions.

- Author(s) should provide concluding statements rather than repetitive statements in the conclusion portion.
    - Conclusion has been revised to not repeat the discussion and provide synthetic insight.

Reviewer 2 Report

  1. Methodology to be enhanced
  2. Add a section of Policy implications

Author Response

Reviewer 2:

We thank the reviewer for their comments and favourable evaluation.
We do not see how a section of Policy implications is applicable.
Unless of course as a downstream application of the extracted sentiment, which is already mentioned in the introduction and related research.
We corrected and improved language issues throughout the manuscript.